# Efficient Sublinear-Regret Algorithms for Online Sparse Linear Regression with Limited Observation

**Shinji Ito**
NEC Corporation
s-ito@me.jp.nec.com

**Daisuke Hatano**
National Institute of Informatics
hatano@nii.ac.jp

**Hanna Sumita**
National Institute of Informatics
sumita@nii.ac.jp

**Akihiro Yabe**
NEC Corporation
a-yabe@cq.jp.nec.com

**Takuro Fukunaga**
JST, PRESTO
takuro@nii.ac.jp

**Naonori Kakimura**
Keio University
kakimura@math.keio.ac.jp

**Ken-ichi Kawarabayashi**
National Institute of Informatics
k-keniti@nii.ac.jp

## Abstract

Online sparse linear regression is the task of applying linear regression analysis to examples arriving sequentially subject to a resource constraint that a limited number of features of examples can be observed. Despite its importance in many practical applications, it has been recently shown that there is no polynomial-time sublinear-regret algorithm unless **NP⊆BPP**, and only an exponential-time sublinear-regret algorithm has been found. In this paper, we introduce mild assumptions to solve the problem. Under these assumptions, we present polynomial-time sublinear-regret algorithms for the online sparse linear regression. In addition, thorough experiments with publicly available data demonstrate that our algorithms outperform other known algorithms.

## 1   Introduction

In online regression, a learner receives examples one by one, and aims to make a good prediction from the features of arriving examples, learning a model in the process. Online regression has attracted attention recently in the research community in managing massive learning data. In real-world scenarios, however, with resource constraints, it is desired to make a prediction with only a limited number of features per example. Such scenarios arise in the context of medical diagnosis of a disease [3] and in generating a ranking of web pages in a search engine, in which it costs to obtain features or only partial features are available in each round. In both these examples, predictions need to be made sequentially because a patient or a search query arrives online.

To resolve the above issue of limited access to features, Kale [7] proposed *online sparse regression*. In this problem, a learner makes a prediction for the labels of examples arriving sequentially over a number of rounds. Each example has $d$ features that can be potentially accessed by the learner. However, in each round, the learner can acquire the values of at most $k'$ features out of the $d$ features, where $k'$ is a parameter set in advance. The learner then makes a prediction for the label of the example. After the prediction, the true label is revealed to the learner, and the learner suffers a loss for making an incorrect prediction. The performance of the prediction is measured here by the standard notion of *regret*, which is the difference between the total loss of the learner and the total

Table 1: Computational complexity of online sparse linear regression.

| Assumptions | | | | Time complexity |
|---|---|---|---|---|
| (1) | (2) | (a) | (b) | |
| ✓ | ✓ | | | Hard [5] |
| ✓ | | ✓ | | **Hard (Theorem 1)** |
| ✓ | ✓ | ✓ | | **Polynomial time (Algorithms 1, 2)** |
| ✓ | ✓ | | ✓ | **Polynomial time (Algorithm 3)** |

loss of the best predictor. In [7], the best predictor is defined as the best $k$-sparse linear predictor, i.e., the label is defined as a linear combination of at most $k$ features.

Online sparse regression is a natural online variant of sparse regression; however, its computational complexity was not well known until recently, as Kale [7] raised a question of whether it is possible to achieve sublinear regret in polynomial time for online sparse linear regression. Foster et al. [5] answered the question by proving that no polynomial-time algorithm achieves sublinear regret unless **NP⊆BPP**. Indeed, this hardness result holds even when observing $\Omega(k \log d)$ features per example. On the positive side, they also proposed an exponential-time algorithm with sublinear regret, when we can observe at least $k + 2$ features in each round. However, their algorithm is not expected to work efficiently in practice. In fact, the algorithm enumerates all the $\binom{d}{k'}$ possibilities to determine $k'$ features in each round, which requires exponential time for any instance.

**Our contributions.**  In this paper, we show that online sparse linear regression admits a polynomial-time algorithm with sublinear regret, under mild practical assumptions. First, we assume that the features of examples arriving online are determined by a hidden distribution (Assumption (1)), and the labels of the examples are determined by a weighted average of $k$ features, where the weights are fixed through all rounds (Assumption (2)). These are natural assumptions in the online linear regression. However, Foster et al. [5] showed that no polynomial-time algorithm can achieve sublinear regret unless **NP⊆BPP** even under these two assumptions.[1]

Owing to this hardness, we introduce two types of conditions on the distribution of features, both of which are closely related to the restricted isometry property (RIP) that has been studied in the literature of sparse recovery. The first condition, which we call *linear independence of features* (Assumption (a)), is stronger than RIP. This condition roughly says that all the features are linearly independent. The second condition, which we call *compatibility* (Assumption (b)), is weaker than RIP. Thus, an instance having RIP always satisfies the compatibility condition. Under these assumptions, we propose the following three algorithms. Here, $T$ is the number of rounds.

- Algorithm 1: A polynomial-time algorithm that achieves $O(\frac{d}{k'-k}\sqrt{T})$ regret, under Assumptions (1), (2), and (a), which requires at least $k + 2$ features to be observed per example.

- Algorithm 2: A polynomial-time algorithm that achieves $O(\sqrt{dT} + \frac{d^{16}}{k'^{16}})$ regret, under Assumptions (1), (2), and (a), which requires at least $k$ features to be observed per example.

- Algorithm 3: A polynomial-time algorithm that achieves $O(\sqrt{dT} + \frac{d^{16}}{k'^{16}})$ regret, under Assumptions (1), (2), and (b), which requires at least $k$ features to be observed per example.

We can also construct an algorithm achieving $O(\frac{d}{k'-k}\sqrt{T})$ regret under Assumption (b) for the case where $k' \geq k + 2$, analogous to Algorithm 1, but we omit it due to space limitations.

Assumptions (1)+(2)+(a) or (1)+(2)+(b) seem to be minimal assumptions needed to achieve sublinear regret in polynomial time. Indeed, as listed in Table 1, the problem is hard if any one of the assumptions is violated, where *hard* means that no polynomial-time algorithm can achieve sublinear regret unless **NP⊆BPP**. Note that Assumption (a) is stronger than (b).

In addition to proving theoretical regret bounds of our algorithms, we perform thorough experiments to evaluate the algorithms. We verified that our algorithms outperform the exponential-time algorithm [5] in terms of computational complexity as well as performance of the prediction. Our algorithms also outperform (baseline) heuristic-based algorithms and algorithms proposed in [2, 6]

for online learning based on limited observation. Moreover, we observe that our algorithms perform well even for a real dataset, which may not satisfy our assumptions (deciding whether the model satisfies our assumptions is difficult; for example, the RIP parameter cannot be approximated within any constant factor under a reasonable complexity assumption [9]). Thus, we can conclude that our algorithm is applicable in practice.

**Overview of our techniques.** One naive strategy for choosing a limited number of features is to choose "large-weight" features in terms of estimated ground-truth regression weights. This strategy, however, does not achieve sublinear regret, as it ignores small-weight features. When we have Assumption (a), we show that if we observe two more features chosen uniformly at random, together with the largest $k$ features, we can make a good prediction. More precisely, using the observed features, we output the label that minimizes the least-square loss function, based on the technique using an unbiased estimator of the gradient [2, 6] and the regularized dual averaging (RDA) method (see, e.g., [11, 4]). This idea gives Algorithm 1, and the details are given in Section 4. The reason why we use RDA is that it is efficient in terms of computational time and memory space as pointed out in [11] and, more importantly, we will combine this with the $\ell_1$ regularization later. However, this requires at least $k + 2$ features to be observed in each round.

To avoid the requirement of two extra observations, the main idea is to employ Algorithm 1 with a partial dataset. As a by-product of Algorithm 1, we can estimate the ground-truth regression weight vector with high probability, even without observing extra features in each round. We use the ground-truth weight vector estimated by Algorithm 1 to choose $k$ features. Combining this idea with RDA adapted for the sparse regression gives Algorithm 2 (Section 5.1) under Assumption (a).

The compatibility condition (Assumption (b)) is often used in LASSO (Least Absolute Shrinkage and Selection Operator), and it is known that minimization with an $\ell_1$ regularizer converges to the sparse solution under the compatibility condition [1]. We introduce $\ell_1$ regularization into Algorithm 1 to estimate the ground-truth regression weight vector when we have Assumption (b) instead of Assumption (a). This gives Algorithm 3 (Section 5.2).

**Related work.** In the online learning problem, a learner aims to predict a model based on the arriving examples. Specifically, in the linear function case, a learner predicts the coefficient $\mathbf{w}_t$ of a linear function $\mathbf{w}_t^\top \mathbf{x}_t$ whenever an example with features $\mathbf{x}_t$ arrives in round $t$. The learner then suffers a loss $\ell_t(\mathbf{w}_t) = (y_t - \mathbf{w}_t^\top \mathbf{x}_t)^2$. The aim is to minimize the total loss $\sum_{t=1}^T (\ell_t(\mathbf{w}_t) - \ell_t(\mathbf{w}))$ for an arbitrary $\mathbf{w}$. It is known that both the gradient descent method [12] and the dual averaging method [11] attain an $O(\sqrt{T})$ regret even for the more general convex function case. However, these methods require access to all features of the examples.

In *linear regression with limited observation*, the limited access to features in regression has been considered [2, 6]. In this problem, a learner can acquire only the values of at most $k'$ features among $d$ features. The purpose here is to estimate a good weight vector, e.g., minimize the loss function $\ell(\mathbf{w})$ or the loss function with $\ell_1$ regularizer $\ell(\mathbf{w}) + \|\mathbf{w}\|_1$. Let us note that, even if we obtain a good weight vector $\mathbf{w}$ with small $\ell(\mathbf{w})$, we cannot always compute $\mathbf{w}^\top \mathbf{x}_t$ from limited observation of $\mathbf{x}_t$ and, hence, in our setting the prediction error might not be as small as $\ell(\mathbf{w})$. Thus, our setting uses a different loss function, defined in Section 2, to minimize the prediction error.

Another problem incorporating the limited access is proposed by Zolghadr et al. [13]. Here, instead of observing $k'$ features, one considers the situation where obtaining a feature has an associated cost. In each round, one chooses a set of features to pay some amount of money, and the purpose is to minimize the sum of the regret and the total cost. They designed an exponential-time algorithm for the problem.

Online sparse linear regression has been studied in [5, 7], but only an exponential-time algorithm has been proposed so far. In fact, Foster et al. [5] suggested designing an efficient algorithm for a special class of the problem as future work. The present paper aims to follow this suggestion.

Recently, Kale et al. [8][2] presented computationally efficient algorithms to achieve sublinear regret under the assumption that input features satisfy RIP. Though this study includes similar results to ours, we can realize some differences. Our paper considers the assumption of the compatibility condition without extra observation (i.e., the case of $k' = k$), whereas Kale et al. [8] studies a

stronger assumption with extra observation ($k' \geq k + 2$) that yields a smaller regret bound than ours. They also studies the agnostic (adversarial) setting.

## 2 Problem setting

**Online sparse linear regression.** We suppose that there are $T$ rounds, and an example arrives online in each round. Each example is represented by $d$ features and is associated with a label, where features and labels are all real numbers. We denote the features of the example arriving in round $t$ by $\mathbf{x}_t = (x_{t1}, \ldots, x_{td})^\top \in \{\mathbf{x} \in \mathbb{R}^d \mid \|\mathbf{x}\| \leq 1\}$, where the norm $\|\cdot\|$ without subscripts denotes the $\ell_2$ norm. The label of each example is denoted by $y_t \in [-1, 1]$.

The purpose of the online sparse regression is to predict the label $y_t \in \mathbb{R}$ from a partial observation of $\mathbf{x}_t$ in each round $t = 1, \ldots, T$. The prediction is made through the following four steps: (i) we choose a set $S_t \subseteq [d] := \{1, \ldots, d\}$ of features to observe, where $|S_t|$ is restricted to be at most $k'$; (ii) observe the selected features $\{x_{ti}\}_{i \in S_t}$; (iii) on the basis of observation $\{x_{ti}\}_{i \in S_t}$, estimate a predictor $\hat{y}_t$ of $y_t$; and (iv) observe the true value of $y_t$.

From $S_t$, we define $D_t \in \mathbb{R}^{d \times d}$ to be the diagonal matrix such that its $(i, i)$th entries are 1 for $i \in S_t$ and the other entries are 0. Then, observing the selected features $\{x_{ti}\}_{i \in S_t}$ in (ii) is equivalent to observing $D_t \mathbf{x}_t$. The predictor $\hat{y}_t$ is computed by $\hat{y}_t = \mathbf{w}_t^\top D_t \mathbf{x}_t$ in (iii).

Throughout the paper, we assume the following conditions, corresponding to Assumptions (1) and (2) in Section 1, respectively.

**Assumption (1)** There exists a weight vector $\mathbf{w}^* \in \mathbb{R}^d$ such that $\|\mathbf{w}\| \leq 1$ and $y_t = \mathbf{w}^{*\top} \mathbf{x}_t + \epsilon_t$ for all $t = 1, \ldots, T$, where $\epsilon_t \sim \mathcal{D}_\epsilon$, independent and identically distributed (i.i.d.), and $\mathbf{E}[\epsilon_t] = 0$, $\mathbf{E}[\epsilon_t^2] = \sigma^2$. There exists a distribution $\mathcal{D}_{\mathbf{x}}$ on $\mathbb{R}^d$ such that $\mathbf{x}_t \sim \mathcal{D}_{\mathbf{x}}$, i.i.d. and independent of $\{\epsilon_t\}$.

**Assumption (2)** The true weight vector $\mathbf{w}^*$ is $k$-sparse, i.e., $S^* = \mathrm{supp}(\mathbf{w}^*) = \{i \in [d] \mid w_i^* \neq 0\}$ satisfies $|S^*| \leq k$.

**Regret.** The performance of the prediction is evaluated based on the *regret* $R_T(\mathbf{w})$ defined by

$$R_T(\mathbf{w}) = \sum_{t=1}^{T} (\hat{y}_t - y_t)^2 - \sum_{t=1}^{T} (\mathbf{w}^\top \mathbf{x}_t - y_t)^2. \tag{1}$$

Our goal is to achieve smaller regret $R_T(\mathbf{w})$ for an arbitrary $\mathbf{w} \in \mathbb{R}^d$ such that $\|\mathbf{w}\| \leq 1$ and $\|\mathbf{w}\|_0 \leq k$. For random inputs and randomized algorithms, we consider the expected regret $\max_{\mathbf{w}: \|\mathbf{w}\|_0 \leq k, \|\mathbf{w}\| \leq 1} \mathbf{E}[R_T(\mathbf{w})]$.

Define the loss function $\ell_t(\mathbf{w}) = (\mathbf{w}^\top \mathbf{x}_t - y_t)^2$. If we compute a predictor $\hat{y}_t = \mathbf{w}_t^\top D_t \mathbf{x_t}$ using a weight vector $\mathbf{w}_t = (w_{t1}, \ldots, w_{td})^\top \in \mathbb{R}^d$ in each step, we can rewrite the regret $R_T(\mathbf{w})$ in (1) using $D_t$ and $\mathbf{w}_t$ as

$$R_T(\mathbf{w}) = \sum_{t=1}^{T} (\ell_t(D_t \mathbf{w}_t) - \ell_t(\mathbf{w})) \tag{2}$$

because $(\hat{y}_t - y_t)^2 = (\mathbf{w}_t^\top D_t \mathbf{x}_t - y_t)^2 = \ell_t(D_t \mathbf{w}_t)$. It is worth noting that if our goal is only to construct $\mathbf{w}_t$ that minimizes the loss function $\ell_t(\mathbf{w}_t)$, then the definition of the regret should be

$$R_T'(\mathbf{w}) = \sum_{t=1}^{T} (\ell_t(\mathbf{w}_t) - \ell_t(\mathbf{w})). \tag{3}$$

However, the goal of online sparse regression involves predicting $\mathbf{y}_t$ from the limited observation. Hence, we use (2) to evaluate the performance. In terms of the regret defined by (3), several algorithms based on limited observation have been developed. For example, the algorithms proposed by Cesa-Bianchi et al. [3] and Hazan and Koren [6] achieve $O(\sqrt{T})$ regret of (3).

# 3 Extra assumptions on features of examples

Foster et al. [5] showed that Assumptions (1) and (2) are not sufficient to achieve sublinear regret. Owing to this observation, we impose extra assumptions.

Let $V := \mathbf{E}[\mathbf{x}_t^\top \mathbf{x}_t] \in \mathbb{R}^{d \times d}$ and let $L$ be the Cholesky decomposition of $V$ (i.e., $V = L^\top L$). Denote the largest and the smallest singular values of $L$ by $\sigma_1$ and $\sigma_d$, respectively. Under Assumption (1) in Section 2, we have $\sigma_1 \leq 1$ because, for arbitrary unit vector $\mathbf{u} \in \mathbb{R}^d$, it holds that $\mathbf{u}^\top V \mathbf{u} = \mathbf{E}[(\mathbf{u}^\top \mathbf{x})^2] \leq 1$. For a vector $\mathbf{w} \in \mathbb{R}^{[d]}$ and $S \subseteq [d]$, we let $\mathbf{w}_S$ denote the restriction of $\mathbf{w}$ onto $S$. For $S \subseteq [d]$, $S^c$ denotes $[d] \setminus S$. We assume either one of the following conditions holds.

(a) **Linear independence of features:** $\sigma_d > 0$.
(b) **Compatibility:** There exists a constant $\phi_0 > 0$ that satisfies $\phi_0^2 \|\mathbf{w}_{S^*}\|_1^2 \leq k \mathbf{w}^\top V \mathbf{w}$ for all $\mathbf{w} \in \mathbb{R}^d$ with $\|\mathbf{w}_{(S^*)^c}\|_1 \leq 2\|\mathbf{w}_{S^*}\|_1$.

We assume the linear independence of features in Sections 4 and 5.1, and the compatibility in Section 5.2 to develop efficient algorithms.

Note that condition (a) means that $L$ is non-singular, and so is $V$. In other words, condition (a) indicates that the features in $\mathbf{x}_t$ are linearly independent. This is the reason why we call condition (a) the "linear independence of features" assumption. Note that the linear independence of features does *not* imply the *stochastic* independence of features.

Conditions (a) and (b) are closely related to RIP. Indeed, condition (b) is a weaker assumption than RIP, and RIP is weaker than condition (a), i.e., (a) linear independence of features $\Longrightarrow$ RIP $\Longrightarrow$ (b) compatibility (see, e.g., [1]). We now clarify how the above two assumptions are connected to the regret. The expectation of the loss function $\ell_t(\mathbf{w})$ is equal to

$$\mathbf{E}_{\mathbf{x}_t, y_t}[\ell_t(\mathbf{w})] = \mathbf{E}_{\mathbf{x}_t \sim \mathcal{D}_\mathbf{x}, \epsilon_t \sim \mathcal{D}_\epsilon}[(\mathbf{w}^\top \mathbf{x}_t - \mathbf{w}^{*\top} \mathbf{x}_t - \epsilon_t)^2]$$
$$= \mathbf{E}_{\mathbf{x}_t \sim \mathcal{D}_\mathbf{x}}[((\mathbf{w} - \mathbf{w}^*)^\top \mathbf{x}_t)^2] + \mathbf{E}_{\epsilon_t \sim \mathcal{D}_\epsilon}[\epsilon_t^\top \epsilon_t] = (\mathbf{w} - \mathbf{w}^*)^\top V(\mathbf{w} - \mathbf{w}^*) + \sigma^2$$

for all $t$, where the second equality comes from $\mathbf{E}[\epsilon_t] = 0$ and that $\mathbf{x}_t$ and $\epsilon_t$ are independent. Denote this function by $\ell(\mathbf{w})$, and then $\ell(\mathbf{w})$ is minimized when $\mathbf{w} = \mathbf{w}^*$. If $D_t$ and $\mathbf{w}_t$ are determined independently of $\mathbf{x}_t$ and $y_t$, the expectation of the regret $R_T(\mathbf{w})$ satisfies

$$\mathbf{E}[R_T(\mathbf{w})] = \mathbf{E}[\sum_{t=1}^{T}(\ell(D_t \mathbf{w}_t) - \ell(\mathbf{w}))] \leq \mathbf{E}[\sum_{t=1}^{T}(\ell(D_t \mathbf{w}_t) - \ell(\mathbf{w}^*))]$$
$$= \mathbf{E}[\sum_{t=1}^{T}(D_t \mathbf{w}_t - \mathbf{w}^*)^\top V(D_t \mathbf{w}_t - \mathbf{w}^*)] = \mathbf{E}[\sum_{t=1}^{T}\|L(D_t \mathbf{w}_t - \mathbf{w}^*)\|^2]. \quad (4)$$

We bound (4) in the analysis.

**Hardness result.** Similarly to [5], we can show that it remains hard under Assumptions (1), (2), and (a). Refer to Appendix A for the proof.

**Theorem 1.** *Let $D$ be any positive constant, and let $c_D \in (0, 1)$ be a constant dependent on $D$. Suppose that Assumptions (1) and (2) hold with $k = O(d^{c_D})$ and $k' = \lfloor kD \ln d \rfloor$. If an algorithm for the online sparse regression problem runs in $\mathrm{poly}(d, T)$ time per iteration and achieves a regret at most $\mathrm{poly}(d, 1/\sigma_d)T^{1-\delta}$ in expectation for some constant $\delta > 0$, then $\mathbf{NP} \subseteq \mathbf{BPP}$.*

# 4 Algorithm with extra observations and linear independence of features

In this section, we present Algorithm 1. Here we assume $k' \geq k + 2$, in addition to the linear independence of features (Assumption (a)). The additional assumption will be removed in Section 5.

As noted in Section 2, our algorithm first computes a weight vector $\mathbf{w}_t$, chooses a set $S_t$ of $k'$ features to be observed, and computes a label $\hat{y}_t$ by $\hat{y}_t = \mathbf{w}_t^\top D_t \mathbf{x}_t$ in each round $t$. In addition, our algorithm constructs an unbiased estimator $\hat{\mathbf{g}}_t$ of the gradient $\mathbf{g}_t$ of the loss function $\ell_t(\mathbf{w})$ at $\mathbf{w} = \mathbf{w}_t$, i.e., $\mathbf{g}_t = \nabla_\mathbf{w}\ell_t(\mathbf{w}_t) = 2\mathbf{x}_t(\mathbf{x}_t^\top \mathbf{w}_t - y_t)$ at the end of the round. In the following, we describe how to compute $\mathbf{w}_t$, $S_t$, and $\hat{\mathbf{g}}_t$ in round $t$, respectively, assuming that $\mathbf{w}_{t'}$, $S_{t'}$, and $\hat{\mathbf{g}}_{t'}$ are computed in the previous rounds $t' = 1, \ldots, t-1$. The entire algorithm is described in Algorithm 1.

**Algorithm 1**

---

**Input:** $\{\mathbf{x}_t, y_t\} \subseteq \mathbb{R}^d \times \mathbb{R}$, $\{\lambda_t\} \subseteq \mathbb{R}_{>0}$, $k' \geq 2$ and $k_1 \geq 0$ such that $k_1 \leq k' - 2$.

1: Set $\hat{\mathbf{h}}_0 = 0$.
2: **for** $t = 1, \ldots, T$ **do**
3:     Define $\mathbf{w}_t$ by (5) and define $S_t$ by $\texttt{Observe}(\mathbf{w}_t, k', k_1)$.
4:     Observe $D_t \mathbf{x}_t$ and output $\hat{y}_t := \mathbf{w}_t^\top D_t \mathbf{x}_t$.
5:     Observe $y_t$ and define $\hat{\mathbf{g}}_t$ by (6) and set $\hat{\mathbf{h}}_t = \hat{\mathbf{h}}_{t-1} + \hat{\mathbf{g}}_t$
6: **end for**

---

**Computing $\mathbf{w}_t$.** We use $\hat{\mathbf{g}}_1, \ldots, \hat{\mathbf{g}}_{t-1}$ to estimate $\mathbf{w}_t$ by the dual averaging method as follows. Define $\hat{\mathbf{h}}_{t-1} = \sum_{j=1}^{t-1} \hat{\mathbf{g}}_j$, which is the average of all estimators of gradients computed in the previous rounds. Moreover, let $(\lambda_1, \ldots, \lambda_T)$ be a monotonically non-decreasing sequence of positive numbers. From these, we define $\mathbf{w}_t$ by

$$\mathbf{w}_t = \underset{\mathbf{w} \in \mathbb{R}^d, \|\mathbf{w}\| \leq 1}{\arg\min} \left\{ \hat{\mathbf{h}}_{t-1}^\top \mathbf{w} + \frac{\lambda_t}{2} \|\mathbf{w}\|^2 \right\} = -\frac{1}{\max\{\lambda_t, \|\hat{\mathbf{h}}_{t-1}\|\}} \hat{\mathbf{h}}_{t-1}, \tag{5}$$

**Computing $S_t$.** Let $k_1$ be an integer such that $k_1 \leq k' - 2$. We define $U_t \subseteq [d]$ as the set of the $k_1$ largest features with respect to $\mathbf{w}_t$, i.e., choose $U_t$ so that $|U_t| = k_1$ and all $i \in U_t$ and $j \in [d] \setminus U_t$ satisfy $|w_{ti}| \geq |w_{tj}|$. Let $V_t$ be the set of $(k' - k_1)$ elements chosen from $[d] \setminus U_t$ uniformly at random. Then our algorithm observes the set $S_t = U_t \cup V_t$ of the $k'$ features. We call this procedure to obtain $S_t$ $\texttt{Observe}(\mathbf{w}_t, k', k_1)$.

**Observation 1.** *We observe that $U_t \subseteq S_t$ and $\mathrm{Prob}[i, j \in S_t] \geq \frac{(k'-k_1)(k'-k_1-1)}{d(d-1)} =: C_{d,k',k_1}$. Thus, $\mathrm{Prob}[i, j \in S_t] > 0$ for all $i, j \in [d]$ if $k' \geq k_1 + 2$.*

For simplicity, we use the notation $p_i^{(t)} = \mathrm{Prob}[i \in S_t]$ and $p_{ij}^{(t)} = \mathrm{Prob}[i, j \in S_t]$ for $i, j \in [d]$.

**Computing $\hat{\mathbf{g}}_t$.** Define $\tilde{X}_t = (\tilde{x}_{tij}) \in \mathbb{R}^{d \times d}$ by $\tilde{X}_t = D_t \mathbf{x}_t^\top \mathbf{x}_t D_t$ and let $X_t \in \mathbb{R}^{d \times d}$ be a matrix whose $(i, j)$-th entry is $\tilde{x}_{tij}/p_{ij}^{(t)}$. It follows that $X_t$ is an unbiased estimator of $\mathbf{x}_t \mathbf{x}_t^\top$. Similarly, defining $\mathbf{z}_t = (z_{ti}) \in \mathbb{R}^d$ by $z_{ti} = x_{ti}/p_i^{(t)}$ for $i \in S_t$ and $z_{ti} = 0$ for $i \notin S_t$, we see that $\mathbf{z}_t$ is an unbiased estimator of $\mathbf{x}_t$. Using $X_t$ and $\mathbf{z}_t$, we define $\hat{\mathbf{g}}_t$ to be

$$\hat{\mathbf{g}}_t = 2X_t \mathbf{w}_t - 2y_t \mathbf{z}_t. \tag{6}$$

**Regret bound of Algorithm 1.** Let us show that the regret achieved by Algorithm 1 is $O(\frac{d}{k'-k} \sqrt{T})$ in expectation.

**Theorem 2.** *Suppose that the linear independence of features is satisfied and $k \leq k' - 2$. Let $k_1$ be an arbitrary integer such that $k \leq k_1 \leq k' - 2$. Then, for arbitrary $\mathbf{w} \in \mathbb{R}^d$ with $\|\mathbf{w}\| \leq 1$, Algorithm 1 achieves $\mathbf{E}[R_T(\mathbf{w})] \leq \frac{3}{\sigma_d^2} \left( \frac{16}{C_{d,k',k_1}} \sum_{t=1}^{T} \frac{1}{\lambda_t} + \frac{\lambda_{T+1}}{2} \right)$. By setting $\lambda_t = 8\sqrt{t}/C_{d,k',k_1}$ for each $t = 1, \ldots, T$, we obtain*

$$\mathbf{E}[R_T(\mathbf{w})] \leq \frac{24}{\sigma_d^2} \sqrt{\frac{d(d-1)}{(k'-k_1)(k'-k_1-1)}} \cdot \sqrt{T+1}. \tag{7}$$

The rest of this section is devoted to proving Theorem 2. By (4), it suffices to evaluate $\mathbf{E}[\sum_{t=1}^{T} \|L(D_t \mathbf{w}_t - \mathbf{w}^*)\|^2]$ instead of $E[R_T(\mathbf{w})]$. The following lemma asserts that each term of (4) can be bounded, assuming the linear independence of features. Proofs of all lemmas are given in the supplementary material.

**Lemma 3.** *Suppose that the linear independence of features is satisfied. If $S_t \supseteq U_t$,*

$$\|L(D_t \mathbf{w}_t - \mathbf{w}^*)\|^2 \leq \frac{3}{\sigma_d^2} \|L(\mathbf{w}_t - \mathbf{w}^*)\|^2. \tag{8}$$

*Proof.* We have

$$\|L(D_t\mathbf{w}_t - \mathbf{w}^*)\|^2 \le \sigma_1^2 \|D_t\mathbf{w}_t - \mathbf{w}^*\|^2 = \sigma_1^2 \left( \sum_{i \in S^* \cap S_t} (w_{ti} - w_i^*)^2 + \sum_{i \in S^* \setminus S_t} w_i^{*2} + \sum_{i \in S_t \setminus S^*} w_{ti}^2 \right)$$

$$\le \sigma_1^2 \left( \|\mathbf{w}_t - \mathbf{w}^*\|^2 + \sum_{i \in S^* \setminus S_t} w_i^{*2} \right), \tag{9}$$

where the second inequality holds since $w_i^* = 0$ for $i \in [d] \setminus S^*$. It holds that

$$\sum_{i \in S^* \setminus S_t} w_i^{*2} \le \sum_{i \in S^* \setminus U_t} w_i^{*2} \le \sum_{i \in S^* \setminus U_t} \left( 2w_{ti}^2 + 2(w_{ti} - w_i^*)^2 \right)$$

$$\le 2 \sum_{i \in U_t \setminus S^*} w_{ti}^2 + 2 \sum_{i \in S^* \setminus U_t} (w_{ti} - w_i^*)^2 \le 2\|\mathbf{w}_t - \mathbf{w}^*\|^2. \tag{10}$$

The first and third inequalities come from $U_t \subseteq S_t$ and the definition of $U_t$. Putting (10) into (9), we have

$$\|L(D_t\mathbf{w}_t - \mathbf{w}^*)\|^2 \le 3\sigma_1^2 \|\mathbf{w}_t - \mathbf{w}^*\|^2 \le \frac{3\sigma_1^2}{\sigma_d^2} \|L(\mathbf{w}_t - \mathbf{w}^*)\|^2.$$

□

It follows from the above lemma that, if $\mathbf{w}_t$ converges to $\mathbf{w}^*$, we have $D_t\mathbf{w}_t = \mathbf{w}^*$, and hence $S_t$ includes the support of $\mathbf{w}^*$. Moreover, it holds that $\sum_{t=1}^T \mathbf{E}[\|L(\mathbf{w}_t - \mathbf{w}^*)\|^2] = \mathbf{E}[\sum_{t=1}^T (\ell_t(\mathbf{w}_t) - \ell_t(\mathbf{w}^*))] = \mathbf{E}[R'_T(\mathbf{w}^*)]$, since $\mathbf{w}_t$ is independent of $\mathbf{x}_t$ and $y_t$. Thus, to bound $\sum_{t=1}^T \mathbf{E}[\|L(\mathbf{w}_t - \mathbf{w}^*)\|^2]$, we shall evaluate $\mathbf{E}[R'_T(\mathbf{w}^*)]$.

**Lemma 4** ([11]). *Suppose that $\mathbf{w}_t$ is defined by (5) for each $t = 1, \ldots, T$, and $\mathbf{w} \in \mathbb{R}^d$ satisfies $\|\mathbf{w}\| \le 1$. Let $G_t = \mathbf{E}[\|\hat{\mathbf{g}}_t\|^2]$ for $t = 1, \ldots, T$. Then,*

$$\mathbf{E}[R'_T(\mathbf{w})] \le \sum_{t=1}^T \frac{1}{\lambda_t} G_t + \frac{\lambda_{T+1}}{2}. \tag{11}$$

If $G_t = O(1)$ and $\lambda_t = \Theta(\sqrt{t})$, the right-hand side of (11) is $O(\sqrt{T})$. The following lemma shows that this is true if $p_{ij}^{(t)} = \Omega(1)$.

**Lemma 5.** *Suppose that the linear independence of features is satisfied. Let $t \in [T]$, and let $q$ be a positive number such that $q \le \min\{p_i^{(t)}, p_{ij}^{(t)}\}$. Then we have $G_t \le 16/q$.*

We are now ready to prove Theorem 2.

*Proof of Theorem 2.* The expectation $\mathbf{E}[R_T(\mathbf{w})]$ of the regret is bounded as $\mathbf{E}[R_T(\mathbf{w})] \le \sum_{t=1}^T \mathbf{E}[\|L(D_t\mathbf{w_t} - \mathbf{w}^*)\|^2] \le \frac{3}{\sigma_d^2} \sum_{t=1}^T \mathbf{E}[\|L(\mathbf{w}_t - \mathbf{w}^*)\|^2] = \frac{3}{\sigma_d^2} \mathbf{E}[R'_T(\mathbf{w}^*)]$, where the first inequality comes from (4) and the second comes from Lemma 3. From Lemma 4, $\mathbf{E}[R'_T(\mathbf{w}^*)]$ is bounded by $\mathbf{E}[R'_T(\mathbf{w}^*)] \le H_T := \sum_{t=1}^T \frac{1}{\lambda_t} G_t + \frac{\lambda_{T+1}}{2}$. Lemma 5 and Observation 1 yield $G_t \le 16/C_{d,k',k_1}$. Hence, for $\lambda_t = 8\sqrt{C_{d,k',k_1}t}$, $H_T$ satisfies $H_T \le \sum_{t=1}^T \frac{16}{C_{d,k',k_1}\lambda_t} + \frac{\lambda_{T+1}}{2} = \sum_{t=1}^T \frac{2}{\sqrt{C_{d,k',k_1}t}} + \frac{4}{\sqrt{C_{d,k',k_1}}}\sqrt{T+1} \le 8\frac{1}{\sqrt{C_{d,k',k_1}}}\sqrt{T+1}$. Combining the above three inequalities, we obtain (7). □

## 5 Algorithms without extra observations

### 5.1 Algorithm 2: Assuming (a) the linear independence of features

In Section 4, Lemma 3 showed a connection between $R_T$ and $R'_T$: $\mathbf{E}[R_T(\mathbf{w})] \le \frac{3\sigma_1^2}{\sigma_d^2}\mathbf{E}[R'_T(\mathbf{w}^*)]$ under $U_t \subseteq S_t$. Then, Lemmas 4 and 5 gave an upper bound of $\mathbf{E}[R'_T(\mathbf{w}^*)]$: $\mathbf{E}[R'_T(\mathbf{w}^*)] = O(\sqrt{T})$

under $p_{ij}^{(t)} = \Omega(1)$. In the case of $k' = k$, however, the conditions $U_t \subseteq S_t$ and $p_{ij}^{(t)} = \Omega(1)$ may not be satisfied simultaneously, since, if $U_t \subseteq S_t$ and $|S_t| = k' = k \geq k_1 = |U_t|$, then we have $U_t = S_t$, which means $p_{ij}^{(t)} = 0$ for $i \notin U_t$ or $j \notin U_t$. Thus, we cannot use both relationships for the analysis. In Algorithm 2, we bound $R_T(\mathbf{w})$ without bounding $R'_T(\mathbf{w})$.

Let us describe an idea of Algorithm 2. To achieve the claimed regret, we first define a subset $J$ of $\{1, 2, \ldots, T\}$ by the set of squares, i.e., $J = \{s^2 \mid s = 1, \ldots, \lfloor \sqrt{T} \rfloor\}$. Let $t_s$ denote the $s$-th smallest number in $J$ for each $s = 1, \ldots, |J|$. In each round $t$, the algorithm computes $S_t$, a weight vector $\tilde{\mathbf{w}}_t$, and a vector $D_t \tilde{\mathbf{g}}_t$, where $\tilde{\mathbf{g}}_t$ is the gradient of $\ell_t(\mathbf{w})$ at $\mathbf{w} = D_t \tilde{\mathbf{w}}_t$. In addition, if $t = t_s$, the algorithm computes other weight vectors $\mathbf{w}_s$ and $\bar{\mathbf{w}}_s := \frac{1}{s} \sum_{j=1}^{s} \mathbf{w}_j$, and an unbiased estimator $\hat{\mathbf{g}}_s$ of the gradient of the loss function $\ell_t(\mathbf{w})$ at $\mathbf{w}_s$.

At the beginning of round $t$, if $t = t_s$, the algorithm first computes $\mathbf{w}_s$, and $\bar{\mathbf{w}}_s$ is defined as the average of $\mathbf{w}_1, \ldots, \mathbf{w}_s$. Roughly speaking, $\mathbf{w}_s$ is the weight vector computed with Algorithm 1 applied to the examples $(x_{t_1}, y_{t_1}), \ldots, (x_{t_s}, y_{t_s})$, setting $k_1$ to be at most $k - 2$. Then, we can show that $\bar{\mathbf{w}}_s$ is a consistent estimator of $\mathbf{w}^*$. This step is only performed if $t \in J$. Then $S_t$ is defined from $\bar{\mathbf{w}}_s$, where $s$ is the largest number such that $t_s \leq t$. Thus, $S_t$ does not change for any $t \in [t_s, t_{s+1} - 1]$. After this, the algorithm computes $\tilde{\mathbf{w}}_t$ from $D_1 \tilde{\mathbf{g}}_1, \ldots, D_{t-1} \tilde{\mathbf{g}}_{t-1}$, and predicts the label of $\mathbf{x}_t$ as $\hat{y}_t := \tilde{\mathbf{w}}_t^\top D_t \mathbf{x}_t$. At the end of the round, the true label $y_t$ is observed, and $D_t \tilde{\mathbf{g}}_t$ is computed from $\mathbf{w}_t$ and $(D_t \mathbf{x}_t, y_t)$. In addition, if $t = t_s$, $\hat{\mathbf{g}}_s$ is computed as in Algorithm 1. We need $\hat{\mathbf{g}}_s$ for computing $\mathbf{w}_{s'}$ with $s' > s$ in the subsequent rounds $t_{s'}$.

The following theorem bounds the regret of Algorithm 2. See the supplementary material for details of the algorithm and the proof of the theorem.

**Theorem 6.** *Suppose that (a), the linear independence of features, is satisfied and $k \leq k'$. Then, there exists a polynomial-time algorithm such that $\mathbf{E}[R_T(\mathbf{w})]$ is at most*

$$8(1+\sqrt{d})\sqrt{T+1} + 12T \sum_{i \in S^*} |w_i^*| \exp\left(-\frac{C_{d,k',0}^2 (T^{\frac{1}{4}} - 1)|w_i^*|^2 \sigma_d^2}{18432}\right) + 4 \sum_{i \in S^*} |w_i^*| \left(\frac{4096}{C_{d,k',0}^2 w_i^{*4} \sigma_d^4} + 1\right)^2,$$

*for arbitrary $\mathbf{w} \in \mathbb{R}^d$ with $\|\mathbf{w}\| \leq 1$, where $C_{d,k',0} = \frac{k'(k'-1)}{d(d-1)} = O(\frac{k'^2}{d^2})$.[2]*

## 5.2 Algorithm 3: Assuming (b) the compatibility condition

Algorithm 3 adopts the same strategy as Algorithm 2 except for the procedure for determining $\mathbf{w}_s$ and $\bar{\mathbf{w}}_s$. In the analysis of Algorithm 2, we show that, to achieve the claimed regret, it suffices to generate $\{S_t\}$ that satisfies $\sum_{t=1}^{T} \text{Prob}[i \notin S_t] = O(\sqrt{T})$ for $i \in S^*$. The condition was satisfied by defining $S_t$ as the set of $k$ largest features with respect to a weight vector $\bar{\mathbf{w}}_s = \sum_{j=1}^{s} \mathbf{w}_j / s$. The linear independence of features guarantees that $\bar{\mathbf{w}}_s$ computed in Algorithm 2 converges to $\mathbf{w}^*$, and hence $\{S_t\}$ defined as above possesses the required property. Unfortunately, if the assumption of the independence of features is not satisfied, e.g., if we have almost same features, then $\bar{\mathbf{w}}_s$ does not converge to $\mathbf{w}^*$. However, if we introduce an $\ell_1$-regularization to the minimization problem in the definition of $\mathbf{w}_s$ and change the definition of $\bar{\mathbf{w}}_s$ to a weighted average of the modified vectors $\mathbf{w}_1, \ldots, \mathbf{w}_s$, then we can generate a required set $\{S_t\}$ under the compatibility assumption. See the supplementary material for details and the proof of the following theorem.

**Theorem 7.** *Suppose that (b), the compatibility assumption, is satisfied and $k \leq k'$. Then, there exists a polynomial-time algorithm such that $\mathbf{E}[R_T(\mathbf{w})]$ is at most*

$$8(1+\sqrt{d})\sqrt{T+1} + 12T \sum_{i \in S^*} |w_i^*| \exp\left(-\frac{C_{d,k',0}\sqrt{T^{\frac{1}{4}} - 1}|w_i^*|^2 \phi_0^2}{5832k}\right) + 4 \sum_{i \in S^*} |w_i^*| \left(\frac{64 \cdot 36^4 k^2}{C_{d,k',0}^2 w_i^{*4} \phi_0^4} + 1\right)^2,$$

*for arbitrary $\mathbf{w} \in \mathbb{R}^d$ with $\|\mathbf{w}\| \leq 1$, where $C_{d,k',0} = \frac{k'(k'-1)}{d(d-1)} = O(\frac{k'^2}{d^2})$.[3],[4]*

# 6 Experiments

In this section, we compare our algorithms with the following four baseline algorithms: (i) a greedy method that chooses the $k'$ largest features with respect to $\mathbf{w}_t$ computed as in Algorithm 1; (ii) a uniform-random method that chooses $k'$ features uniformly at random; (iii) the algorithm of [6] (called AELR); and (iv) the algorithm of [5] (called FKK). Owing to space limitations, we only present typical results here. Other results and the detailed descriptions on experiment settings are provided in the supplementary material.

**Synthetic data.** First we show results on two kinds of synthetic datasets: instances with $(d, k, k')$ and instances with $(d, k_1, k)$. We set $k_1 = k$ in the setting of $(d, k, k')$ and $k' = k$ in the setting of $(d, k_1, k)$. The instances with $(d, k, k')$ assume that Algorithm 1 can use the ground truth $k$, while Algorithm 1 cannot use $k$ in the instances with $(d, k_1, k)$. For each $(d, k, k')$ and $(d, k_1, k)$, we executed all algorithms on five instances with $T = 5000$ and computed the averages of regrets and run time, respectively. When $(d, k, k') = (20, 5, 7)$, FKK spent 1176 s on average, while AELR spent 6 s, and the others spent at most 1 s.

Figures 1 and 2 plot the regrets given by (1) over the number of rounds on a typical instance with $(d, k, k') = (20, 5, 7)$. Tables 2 and 3 summarize the average regrets at $T = 5000$, where A1, A2, A3, G, and U denote Algorithm 1, 2, 3, greedy, and uniform random, respectively. We observe that Algorithm 1 achieves smallest regrets in the setting of $(d, k, k')$, whereas Algorithms 2 and 3 are better than Algorithm 1 in the setting of $(d, k_1, k)$. The results match our theoretical results.

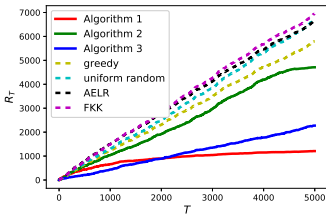
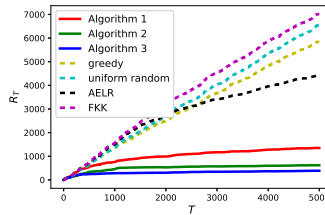
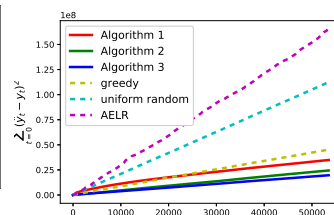

Figure 1: Plot of regrets with $(d, k, k') = (20, 5, 7)$

Figure 2: Plot of regrets with $(d, k_1, k) = (20, 5, 7)$

Figure 3: CT-slice datasets

Table 2: Values of $R_T/10^2$ when changing $(d, k, k')$.

Table 3: Values of $R_T/10^2$ when changing $(d, k_1, k)$.

| $(d, k_1, k)$ | A1 | A2 | A3 | G | U | AELR | FKK |
|---|---|---|---|---|---|---|---|
| (10,2,4) | **1.53** | 2.38 | 3.60 | 33.28 | 25.73 | 60.76 | 24.05 |

| $(d, k_1, k)$ | A1 | A2 | A3 | G | U | AELR | FKK |
|---|---|---|---|---|---|---|---|
| (10,2,4) | 26.88 | 20.59 | **17.19** | 43.03 | 60.02 | 64.75 | 58.71 |

**Real data.** We next conducted experiments using a CT-slice dataset, which is available online [10]. Each data consists of 384 features retrieved from 53500 CT images associated with a label that denotes the relative position of an image on the axial axis.

We executed all algorithms except FKK, which does not work due to its expensive run time. Since we do not know the ground-truth regression weights, we measure the performance by the first term of (1), i.e., square loss of predictions. Figure 3 plots the losses over the number of rounds. The parameters are $k_1 = 60$ and $k' = 70$. For this instance, the run times of Algorithms 1 and 2, greedy, uniform random, and AELR were 195, 35, 147, 382, and 477 s, respectively.

We observe that Algorithms 2 and 3 are superior to the others, which implies that Algorithm 2 and 3 are suitable for instances where the ground truth $k$ is not known, such as real data-based instances.

## Acknowledgement

This work was supported by JST ERATO Grant Number JPMJER1201, Japan.

## Footnotes

[1] Although the statement in [5] does not mention the assumptions, its proof indicates that the hardness holds even with these assumptions.

[2]The paper [8] was published after our manuscript was submitted.

[3] The asymptotic regret bound mentioned in Section 1, can be yielded by bounding the second term with the aid of the following: $\max_{T \geq 0} T \exp(-\alpha T^\beta) = (\alpha\beta)^{-\frac{1}{\beta}} \exp(-1/\beta)$ for arbitrary $\alpha > 0$, $\beta > 0$.

[4] Note that $\phi_0$ is the constant appearing in Assumption (b) in Section 3.

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
