[Supplementary Material · main_proofread.pdf]

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

# Appendix

## A  Proof of Theorem 1

As is the case for Theorem 2 in [6], our reduction starts from the work of Dinur and Steurer [5].

**Theorem 8** ([5]). *For any given constant $D > 0$, there is a constant $c_D \in (0, 1)$ and a $\mathrm{poly}(n^D)$-time algorithm that takes a **3CNF** formula $\phi$ of size $n$ as input and constructs a **Set Cover** instance over a ground set of size $m = \mathrm{poly}(n^D)$ with $d = \mathrm{poly}(n)$ sets, with the following properties:*

1. *if $\phi \in$ **SAT**, then there is a collection of $k = O(d^{c_D})$ sets, which covers each element exactly once; and*

2. *if $\phi \notin$ **SAT**, then no collection of $k' = \lfloor D \ln(d) k \rfloor < d$ sets covers all elements; i.e., at least one element is left uncovered.*

The **Set Cover** instance generated from $\phi$ can be encoded as a binary matrix $M_\phi \in \{0, 1\}^{m \times d}$ with the rows corresponding to the elements of the ground set, and the columns corresponding to the sets such that each column is the indicator vector of the corresponding set. From the definition of $M_\phi$ and the above theorem, if $\phi \in$ **SAT**, then there exists a $k$-sparse binary vector $\mathbf{z} \in \{0, 1\}^m$ $M_\phi \mathbf{z} = \mathbf{1}$, where $\mathbf{1}$ is the all-ones vector, and if $\phi \notin$ **SAT**, then for arbitrary $S \subseteq [d]$ such that $|S| \leq k'$, there exists at least one row of $M_\phi$ that is 0 in all the coordinates in $S$.

Using this reduction, we show that an algorithm **Alg** for online sparse algorithm with properties of Theorem 1 can be used to give a **BPP** algorithm for **SAT**. Algorithm 4 is a randomized algorithm for deciding satisfiability of a given **3CNF** formula $\phi$ using the algorithm **Alg**. Since Step 3 runs in polynomial time and since $T$ is a polynomial in $n$ and **Alg** runs in $\mathrm{poly}(d, T)$ time per iteration, Algorithm 4 is a polynomial-time algorithm.

We now claim that this algorithm correctly decides satisfiability of $\phi$ with probability at least $3/4$, and is hence a **BPP** algorithm for **SAT**.

Suppose that $\phi \in$ **SAT**. Then there exists a $k$-sparse vector $\mathbf{z} \in \{0, 1\}^d$ such that $M_\phi \mathbf{z} = \mathbf{1}_m$. Hence, for $X$ and $\mathbf{y}$ defined in Algorithm 4, we have

$$
\begin{aligned}
\gamma = \min_{\mathbf{w} \in \mathbb{R}^d} \|X\mathbf{w} - \tilde{\mathbf{y}}\|_2^2 &\leq \min_{\mathbf{w} \in \mathbb{R}^d, \|\mathbf{w}\|_0 \leq k'} \|X\mathbf{w} - \tilde{\mathbf{y}}\|_2^2 \\
&\leq \|\frac{1}{\sqrt{d}} X\mathbf{z} - \tilde{\mathbf{y}}\|_2^2 = \|\frac{1}{4\sqrt{d}(m+d)^3} \mathbf{z}\|_2^2 \leq \frac{1}{16(m+d)^6},
\end{aligned}
\tag{13}
$$

which means that Step 5 is not executed. Since the $\ell_2$ norm of each row of $X$ and each entry of $\mathbf{y}$ are at most 1, it holds that $\|\mathbf{x}_t\| \leq 1$ and $|y_t| \leq 1$. Next, let us see that $\mathbf{x}_t, y_t$ satisfies Assumptions (1), (2), and (a). Since $\mathbf{y} = X\tilde{\mathbf{w}}$, it holds for all $t$ that $y_t = \tilde{\mathbf{w}}^\top x_t$, which means that Assumption (1) holds, where $\epsilon_t = 0$. From the definition of $\mathbf{x}_t$, $\mathbf{x}_1, \ldots, \mathbf{x}_T$ follows a distribution on $\mathbb{R}^d$ independently, and independent of $\epsilon_t$ because $\epsilon_t$ are constant, and, hence, Assumption (2) holds. Moreover, $V = \mathbf{E}[\mathbf{x}_t \mathbf{x}_t^\top] = \frac{1}{m+d} X^\top X$ is non-singular (i.e., Assumption (a) holds) and has the smallest singular value at least $\frac{1}{(m+d)}$. Indeed, for an arbitrary $d$-dimensional unit vector $\mathbf{u} \in \mathbb{R}^d$, the $\ell_2$ norm of $X\mathbf{u} = [\frac{1}{\sqrt{d}} M_\phi \mathbf{u}; \frac{1}{4(m+d)^3} \mathbf{u}]$ is at least $\frac{1}{4(m+d)^3}$ and, hence, $\mathbf{u}^\top X^\top X \mathbf{u} \geq \frac{1}{(4(m+d)^3)^2}$, which means that $\sigma_d$, the smallest singular value of $V = \frac{1}{m+d} X^\top X$, is at least $\frac{1}{16(m+d)^7}$. Let us now show that $\mathbf{E}[\sum_{t=0}^T (y_t - \hat{y}_t)^2] \leq \frac{T}{8(m+d)^5}$. From the assumption on **Alg**, it holds for all $k$-sparse vectors $\mathbf{w}$ that

$$
\mathbf{E}[R_T(\mathbf{w})] \leq p_2(d, 16(m+d)^7) T^{1-\delta} \leq \frac{T}{16(m+d)^5},
\tag{14}
$$

---

**Algorithm 4** An algorithm for deciding satisfiability of 3CNF formula

---

**Input:** A constant $D > 0$, and an algorithm **Alg** for the $(k, k', d)$-online sparse regression problem, where $k, k', d$ and $c_D$ are the constants from Theorem 8, that runs in $p_1(d, T)$ time per iteration, with expected regret bounded by $p_2(d, 1/\sigma_d)T^{1-\delta}$ under Assumptions (1), (2), and (4) in Section 2 and Assumption (a) in Section 3. A **3CNF** formula $\phi$.

1: Compute the matrix $M_\phi$ and the associated parameters $k, k', d, m$ from Theorem 8.
2: Define $X \in \mathbb{R}^{(m+d) \times d}$ and $\tilde{\mathbf{y}} \in \mathbb{R}^{m+d}$ by

$$X = \begin{bmatrix} \frac{1}{\sqrt{d}} M_\phi \\ \frac{1}{4(m+d)^3} I_d \end{bmatrix}, \quad \tilde{\mathbf{y}} = \begin{bmatrix} \frac{1}{d} \mathbf{1}_m \\ \mathbf{0}_d \end{bmatrix}. \tag{12}$$

3: Compute $\tilde{\mathbf{w}} \in \arg\min_{\mathbf{w} \in \mathbb{R}^m} \|X\mathbf{w} - \tilde{\mathbf{y}}\|_2^2$ and define $\mathbf{y} = X\tilde{\mathbf{w}}, \gamma = \|X\tilde{\mathbf{w}} - \tilde{\mathbf{y}}\|_2^2$.
4: **if** $\gamma > \frac{1}{16d^2}$ **then**
5:   Return "unsatisfiable."
6: **end if**
7: Run **Alg** with the parameters $k, k', d$ for $T := \lceil \max\{(16(m + d)^5 p_1(d, 16(m + d)^7))^{1/\delta}, 256(m + d)^{10}\} \rceil$ iterations.
8: **for** $t = 1, \ldots, T$ **do**
9:   Sample $i$ from $[m + d]$ uniformly at random and set $\mathbf{x}_t$ and $y_t$ to be the $i$-th row of $X$ and the $i$-th entry of $\mathbf{y}$, respectively.
10:   Obtain a set of coordinates $S_t$ of size at most $k'$ by running **Alg**, and provide it with the coordinates $\mathbf{x}_t(S_t)$.
11:   Obtain the prediction $\hat{y}_t$ from **Alg**, and provide it with the true label $y_t$.
12: **end for**
13: **if** $\sum_{t=1}^T (y_t - \hat{y}_t)^2 \leq \frac{T}{2(m+d)^5}$ **then**
14:   Return "satisfiable."
15: **else**
16:   Return "unsatisfiable."
17: **end if**

---

where the second inequality comes from $T \geq (16(m + d)^5 p_1(d, 16(m + d)^7))^{\frac{1}{\delta}}$. Since $\mathbf{z}$ is a $k$-sparse binary vector, we have

$$\mathbf{E}[R_T(\frac{1}{\sqrt{d}}\mathbf{z})] = \mathbf{E}[\sum_{t=1}^T (y_t - \hat{y}_t)^2] - \mathbf{E}[\sum_{t=1}^T (y_t - \frac{1}{\sqrt{d}}\mathbf{z}^\top \mathbf{x}_t)^2]$$

$$= \mathbf{E}[\sum_{t=1}^T (y_t - \hat{y}_t)^2] - \frac{T}{m + d}\|\mathbf{y} - \frac{1}{\sqrt{d}}X\mathbf{z}\|_2^2.$$

Since we have $\|\mathbf{y} - \frac{1}{\sqrt{d}}X\mathbf{z}\|_2^2 \leq \|\tilde{\mathbf{y}} - \frac{1}{\sqrt{d}}X\mathbf{z}\|_2^2 \leq \frac{1}{16(m+d)^6}$, we obtain

$$\mathbf{E}[\sum_{t=1}^T (y_t - \tilde{y}_t)^2] \leq \frac{T}{16(m + d)^5} + \frac{T}{16(m + d)^7} \leq \frac{T}{8(m + d)^5}. \tag{15}$$

Since $\sum_{t=1}^T (y_t - \tilde{y}_t)^2$ is a non-negative random variable, by Markov's inequality we conclude that with probability at least $3/4$, the total loss $\sum_{t=1}^T (y_t - \tilde{y}_t)^2$ is bounded by $\frac{T}{2(m+d)^5}$ from above, and hence Algorithm 4 correctly outputs "satisfiable."

Next, suppose $\phi \notin$ **SAT**. If $\gamma > \frac{1}{16d^2}$, then Algorithm 4 correctly outputs "unsatisfiable." Hence, it suffices to consider the case of $\gamma \leq \frac{1}{16d^2}$. Fix any round $t$ and let $S_t$ be the set of coordinates of size at most $k'$ selected by **Alg** to query. Since $\phi \notin$ **SAT**, there is at least one element in the ground set that is not covered by any set among these $k'$ sets. This implies that there is at least one row of $M_\phi$ that is 0 in all the coordinates in $S_t$. Let $d_1$ denote the number of such rows. Further, the number $d_2$ of rows of $I_d$ that are 0 in all the coordinates in $S_t$ is equal to $d - k'$. Since $\mathbf{x}_t$ is a uniformly random

row of $X$ chosen independently of $S_t$, we have

$$\mathrm{Prob}[\mathbf{x}_t(S_t) = \mathbf{0}] = \frac{d_1 + d_2}{m + d} \geq \frac{2}{m + d}. \tag{16}$$

The conditional probability that given $\mathbf{x}_t(S_t) = \mathbf{0}$, $i$ in Step 9 is at most $m$ is equal to $\frac{d_1}{d_1 + d_2}$.

Now, we claim that $\mathbf{E}[(y_t - \hat{y}_t)^2 \mid \mathbf{x}_t(S_t) = \mathbf{0}] \geq \frac{1}{2(m+d)^4}$. Since $y_t$ and $\hat{y}_t$ are conditionally independent given $\mathbf{x}_t(S_t)$, we have $\mathbf{E}[(y_t - \hat{y}_t)^2 \mid \mathbf{x}_t(S_t) = \mathbf{0}] \geq \mathrm{var}[y_t \mid \mathbf{x}_t(S_t) = \mathbf{0}]$. Let us recall that $\mathrm{Prob}[i \leq m \mid \mathbf{x}_t(S_t) = \mathbf{0}] = \frac{d_1}{d_1 + d_2}$ for $i$ in Step 9. If $i \leq m$, then $y_t \geq \frac{3}{4d}$ and otherwise $y_t \leq \frac{1}{4d}$ since the difference between $\mathbf{y}$ and $\tilde{\mathbf{y}}$ is bounded in absolute value by $\|\mathbf{y} - \tilde{\mathbf{y}}\|_2 \leq \sqrt{\gamma} \leq 1/3$ and the $i$th element of $\tilde{\mathbf{y}}$ is 1 if $i \leq m$ and 0 otherwise. Hence, given $\mathbf{x}_t(S_t) = \mathbf{0}$, $y_t \geq \frac{3}{4d}$ with probability $\frac{d_1}{d_1 + d_2}$ and $y_t \leq \frac{1}{4d}$ with probability $\frac{d_2}{d_1 + d_2}$. Since $d_1, d_2 \geq 1$ and $d_1 + d_2 \leq m + d$, we have $\mathbf{E}[(y_t - \hat{y}_t)^2 \mid \mathbf{x}_t(S_t) = \mathbf{0}] \geq \frac{2}{(m+d)^2} \cdot \frac{1}{(2d)^2} \geq \frac{1}{2(m+d)^4}$. Further, from (16), we obtain

$$\mathbf{E}[(y_t - \hat{y}_t)^2] \geq \mathbf{E}[(y_t - \hat{y}_t)^2 \mid \mathbf{x}_t(S_t) = \mathbf{0}] \cdot \mathrm{Prob}[\mathbf{x}_t(S_t) = \mathbf{0}] \geq \frac{1}{2(m+d)^4} \cdot \frac{2}{m+d} = \frac{1}{(m+d)^5}. \tag{17}$$

Let $\mathbf{E_t}$ denote the expectation of a random variable conditioned on all randomness prior to round $t$. Since the choices of $\mathbf{x}_t$ and $y_t$ are independent of previous choices in each round, the same argument also implies that $\mathbf{E}_t[(y_t - \hat{y}_t)^2] \geq \frac{1}{(m+d)^5}$. Applying Azuma's inequality to the martingale difference sequence $\mathbf{E}_t[(y_t - \hat{y}_t)^2] - (y_t - \hat{y}_t)^2$ for $t = 1, \ldots, T$, since each term is bounded in absolute value by 4, we obtain

$$\mathrm{Prob}[\sum_{t=1}^{T} \mathbf{E}_t[(y_t - \hat{y}_t)^2] - (y_t - \hat{y}_t)^2 \geq 8\sqrt{T}] \leq \exp(-\frac{64T}{2 \cdot 16T}) \leq \frac{1}{4}. \tag{18}$$

Thus, with probability at least $3/4$, the total loss $\sum_{t=1}^{T}(y_t - \hat{y}_t)^2$ is greater than $\sum_{t=1}^{T} \mathbf{E}_t[(y_t - \hat{y}_t)^2] - 8\sqrt{T} \geq \frac{T}{(m+d)^5} - 8\sqrt{T} \geq \frac{T}{2(m+d)^5}$ since $T \geq 256(m+d)^{10}$. Thus, in the case of $\phi \notin \textbf{SAT}$, Algorithm 4 correctly outputs "unsatisfiable" with probability at least $3/4$.

## B  Preliminary lemmas

Before presenting the statement, let us introduce some notation. In the following, we use the following facts without notice:

- $\|\mathbf{g}_t\| = 2\|\mathbf{x}_t(\mathbf{x}_t^\top \mathbf{w}_t - y_t)\|$ is bounded from above by 4; and
- $|\ell_t(\mathbf{w}) - \ell(\mathbf{w})|$ is bounded from above by 8;

which come from $\mathbf{w} \leq 1$, $\|\mathbf{x}_t\| \leq 1$ and $|y_t| \leq 1$. Further, we define the following numbers for notational convenience:

- $G = 4/C_{d,k',k_1}$, an upper bound of $\|\hat{\mathbf{g}}_t - \mathbf{g}_t\|$;
- $C = 128/C_{d,k',k_1}^2 = 8G^2$.

The following lemma is used for bounding $R'_T(\mathbf{w})$.

**Lemma 9.** *Let* $(\lambda_1, \ldots, \lambda_T)$ *be a monotonically non-decreasing sequence of positive numbers, and let* $(\zeta_1, \ldots, \zeta_T)$ *be a sequence of non-negative numbers. If* $\mathbf{w}_t$ *is defined by*

$$\mathbf{w}_t = \underset{\mathbf{w} \in \mathbb{R}^d, \|\mathbf{w}\| \leq 1}{\arg\min} \{\hat{\mathbf{h}}_t^\top \mathbf{w} + \frac{\lambda_t}{2}\|\mathbf{w}\|^2 + \sum_{j=1}^{t} \zeta_j \|\mathbf{w}\|_1\}$$

*for all* $t = 1, \ldots, T$, *then, for any* $\mathbf{w} \in \mathbb{R}^d$ *satisfying* $\|\mathbf{w}\| \leq 1$, *it holds that*

$$\sum_{t=1}^{T} \left( \hat{\mathbf{g}}_t^\top (\mathbf{w}_t - \mathbf{w}) + \zeta_t(\|\mathbf{w}_t\|_1 - \|\mathbf{w}\|_1) \right) \leq \sum_{t=1}^{T} \frac{1}{\lambda_t}\|\hat{\mathbf{g}}_t\|^2 + \frac{\lambda_{T+1}}{2}.$$

*Proof.* See, e.g., [12]. □

The following lemma is used for bounding the gap between $\sum_{t=1}^{T} \|L(\mathbf{w}_t - \mathbf{w}^*)\|^2$ and $\sum_{t=1}^{T} \hat{\mathbf{g}}_t^\top (\mathbf{w}_t - \mathbf{w})$, with high probability.

**Lemma 10.** *For arbitrary $\delta > 0$, we have*

$$\text{Prob}[\sum_{t=1}^{T} \|L(\mathbf{w}_t - \mathbf{w}^*)\|^2 \geq \sum_{t=1}^{T} \hat{\mathbf{g}}_t^\top (\mathbf{w}_t - \mathbf{w}) + 3\delta] \leq 3 \exp\left(-\frac{\delta^2}{Ct}\right). \tag{19}$$

*Proof.* We have $\sum_{j=1}^{t} \|L(\mathbf{w}_j - \mathbf{w}^*)\|^2 = \sum_{j=1}^{t} (\ell(\mathbf{w}_j) - \ell(\mathbf{w}^*))$, and this can be expanded as

$$\sum_{j=1}^{t} (\ell(\mathbf{w}_j) - \ell(\mathbf{w}^*)) = \sum_{j=1}^{t} (\ell(\mathbf{w}_j) - \ell_j(\mathbf{w}_j)) + \sum_{j=1}^{t} (\ell_j(\mathbf{w}^*) - \ell(\mathbf{w}^*)) + \sum_{j=1}^{t} (\ell_j(\mathbf{w}_j) - \ell_j(\mathbf{w}^*)).$$

From the convexity of $\ell_j$, the term $\sum_{j=1}^{t} (\ell_j(\mathbf{w}_j) - \ell_j(\mathbf{w}^*))$ can be bounded as

$$\sum_{j=1}^{t} (\ell_j(\mathbf{w}_j) - \ell_j(\mathbf{w}^*)) \leq \sum_{j=1}^{t} \mathbf{g}_j^\top (\mathbf{w}_j - \mathbf{w}^*) = \sum_{j=1}^{t} \hat{\mathbf{g}}_j^\top (\mathbf{w}_j - \mathbf{w}^*) + \sum_{j=1}^{t} (\mathbf{g}_j - \hat{\mathbf{g}}_j)^\top (\mathbf{w}_j - \mathbf{w}^*).$$

Summarizing the above inequalities, $\sum_{j=1}^{t} \|L(\mathbf{w}_j - \mathbf{w}^*)\|^2 - \sum_{t=1}^{T} \hat{\mathbf{g}}_t^\top (\mathbf{w}_t - \mathbf{w})$ is bounded from above by the sum of (i) $\sum_{j=1}^{t} (\mathbf{g}_j - \hat{\mathbf{g}}_j)^\top (\mathbf{w}_j - \mathbf{w}^*)$, (ii) $\sum_{j=1}^{t} (\ell(\mathbf{w}_j) - \ell_j(\mathbf{w}_j))$, and (iii) $\sum_{j=1}^{t} (\ell_j(\mathbf{w}^*) - \ell(\mathbf{w}^*))$. Let us construct bounds for each term.

First, consider (i). Denote the $j$-th input data $(\mathbf{x}_j, y_j)$ by $z_j$. Since we have

$$\mathbf{E}[(\mathbf{g}_j - \hat{\mathbf{g}}_j)(\mathbf{w}_j - \mathbf{w}^*)|z_1, \dots, z_{j-1}, S_1, \dots, S_{j-1}] = 0$$

and $|(\mathbf{g}_j - \hat{\mathbf{g}}_j)^\top (\mathbf{w}_j - \mathbf{w}^*)| \leq 2G$, we can apply the Azuma–Hoeffding inequality to (i) to obtain that

$$\text{Prob}[\sum_{j=1}^{t} (\mathbf{g}_j - \hat{\mathbf{g}}_j)^\top (\mathbf{w}_j - \mathbf{w}^*) \geq \delta] \leq \exp(\frac{-\delta^2}{8tG^2}).$$

The value of (ii) also can be bounded by using the Azuma–Hoeffding inequality, since we have

$$\mathbf{E}[\ell(\mathbf{w}_j) - \ell_j(\mathbf{w}_j)|z_1, \dots, z_{j-1}, S_1, \dots, S_{j-1}] = 0,$$

and $|\ell(\mathbf{w}_j) - \ell_j(\mathbf{w}_j)| \leq 8$. Accordingly, we obtain

$$\text{Prob}[\sum_{j=1}^{t} \ell(\mathbf{w}_j) - \ell_j(\mathbf{w}_j) \geq \delta] \leq \exp(\frac{-\delta^2}{128t}).$$

Similarly, since $\{\ell_j(\mathbf{w}^*) - \ell(\mathbf{w}^*)\}$ are independent random variables such that $\mathbf{E}[\ell_j(\mathbf{w}^*) - \ell(\mathbf{w}^*)] = 0$ and $|\ell_j(\mathbf{w}^*) - \ell(\mathbf{w}^*)| \leq 8$, the value of (iii) can be bounded by using Hoeffding's inequality, as follows:

$$\text{Prob}[\sum_{j=1}^{t} \ell_j(\mathbf{w}^*) - \ell(\mathbf{w}^*) \geq \delta] \leq \exp(\frac{-\delta^2}{128t}).$$

Summarizing the above inequalities, we have

$$\text{Prob}\left[\sum_{j=1}^{t} \|L(\mathbf{w}_j - \mathbf{w}^*)\|^2 \geq \sum_{t=1}^{T} \hat{\mathbf{g}}_t^\top (\mathbf{w}_t - \mathbf{w}) + 3\delta\right] \leq 2 \exp(\frac{-\delta^2}{128t}) + \exp(\frac{-\delta^2}{8tG^2}) \leq 3 \exp(\frac{-\delta^2}{Ct}).$$

By substituting this for $\lambda_j = 2G\sqrt{j}$, we obtain (19). □

# C   Proofs of the lemmas in Section 4

## C.1   Proof of Lemma 4

Since $\ell_t(\mathbf{w})$ is a convex function, it holds for arbitrary $\mathbf{w} \in \mathbb{R}^d$ that $\ell_t(\mathbf{w}_t) - \ell_t(\mathbf{w}) \leq \mathbf{g}_t^\top(\mathbf{w}_t - \mathbf{w})$. By taking the expectation, we have $\mathbf{E}[\ell_t(\mathbf{w}_t) - \ell_t(\mathbf{w})] \leq \mathbf{E}[\mathbf{g}_t^\top(\mathbf{w}_t - \mathbf{w})] = \mathbf{E}[\hat{\mathbf{g}}_t^\top(\mathbf{w}_t - \mathbf{w})]$ since $\hat{\mathbf{g}}_t$ is an unbiased estimator of $\mathbf{g}_t$. By taking the sum of this inequality for $t = 1, \ldots, T$, we obtain that

$$\mathbf{E}[R_T'(\mathbf{w}^*)] \leq \mathbf{E}[\sum_{t=1}^T \hat{\mathbf{g}}_t^\top(\mathbf{w}_t - \mathbf{w}^*)].$$

From Lemma 9, the right-hand side of this can be bounded as

$$\mathbf{E}[\sum_{t=1}^T \hat{\mathbf{g}}_t^\top(\mathbf{w}_t - \mathbf{w}^*)] \leq \mathbf{E}[\sum_{t=1}^T \frac{1}{\lambda_t}\|\hat{\mathbf{g}}_t\|^2 + \frac{\lambda_{T+1}}{2}] \leq \sum_{t=1}^T \frac{1}{\lambda_t}G_t + \frac{\lambda_{T+1}}{2},$$

where we used $\mathbf{E}[\|\mathbf{g}_t\|^2] = G_t$ in the second inequality.

## C.2   Proof of Lemma 5

*Proof.* First, $\hat{\mathbf{g}}_t$ defined by (6) satisfies

$$\|\hat{\mathbf{g}}_t\|^2 = \|2X\mathbf{w}_t - 2y_t\mathbf{z}\|^2 \leq 2\|2X\mathbf{w}_t\|^2 + 2\|2y_t\mathbf{z}\|^2 \leq 8\|X\|_{\mathrm{F}}^2\|\mathbf{w}_t\|^2 + 8|y_t|\|\mathbf{z}\|^2.$$

The expectation of $\|X\|_{\mathrm{F}}^2$ is bounded as

$$\mathbf{E}[\|X\|_{\mathrm{F}}^2] = \mathbf{E}\left[\sum_{1 \leq i,j \leq d} p_{ij}^{(t)}(\frac{x_{ti}x_{tj}}{p_{ij}^{(t)}})^2\right] \leq \mathbf{E}\left[\frac{1}{q}\sum_{1 \leq i,j \leq d}(x_ix_j)^2\right] \leq \frac{1}{q}.$$

Similarly, we have $\mathbf{E}[\|\mathbf{z}\|^2] \leq 1/q$. By combining these inequalities, we obtain Lemma 5.   $\square$

# D   Details of Algorithm 2

**Estimating $\mathbf{w}^*$**   Although we assumed $k' \geq k + 2$ in the analysis of the regret bound, Algorithm 1 can be defined even if $k' = k$ by setting $k_1$ to a number at most $k$. At the end of round $t$, Algorithm 1 keeps weight vectors $\mathbf{w}_1, \ldots, \mathbf{w}_t$. From these weight vectors, define $\bar{\mathbf{w}}_t$ as $\frac{1}{t}\sum_{j=1}^t \mathbf{w}_j$. In the following, we prove that $\bar{\mathbf{w}}_t$ is a consistent estimator of $\mathbf{w}^*$ even if $k' = k$ and $k_1 \leq k - 2$. We use this fact in the next section.

**Proposition 11.** *Let $\bar{\mathbf{w}}_t$ be the average of $\mathbf{w}_1, \ldots, \mathbf{w}_t$ computed by Algorithm 1 with setting $\lambda_t = \frac{8}{C}\sqrt{t}$ for $t = 1, \ldots, T$. Then, for arbitrary $\delta > 0$, $\|L(\bar{\mathbf{w}}_t - \mathbf{w}^*)\|^2 \leq \frac{1}{t}(\frac{8}{C}\sqrt{t+1} + 3\delta)$ holds with probability at least $1 - 3\exp(\frac{-C^2\delta^2}{128t})$. Accordingly, assuming the linear independence of features, $\|\bar{\mathbf{w}}_t - \mathbf{w}^*\|^2 \leq \frac{1}{t\sigma_d^2}(\frac{8}{C}\sqrt{t+1} + 3\delta)$ holds with probability at least $1 - 3\exp(\frac{-C^2\delta^2}{128t})$.*

*Proof.* From the convexity of the square loss and Jensen's inequality, we have $\|L(\bar{\mathbf{w}}_t - \mathbf{w}^*)\| \leq \frac{1}{t}\sum_{j=1}^t \|L(\mathbf{w}_j - \mathbf{w}^*)\|$. Hence, it suffices to show that

$$\text{Prob}\left[\sum_{j=1}^t \|L(\mathbf{w}_j - \mathbf{w}^*)\| \geq 2G\sqrt{t+1} + 3\delta\right] \leq 3\exp(\frac{-\delta^2}{Ct}). \tag{20}$$

We have $\sum_{j=1}^t \|L(\mathbf{w}_j - \mathbf{w}^*)\|^2 = \sum_{j=1}^t (\ell(\mathbf{w}_j) - \ell(\mathbf{w}^*))$ and this can be expanded as

$$\sum_{j=1}^t (\ell(\mathbf{w}_j) - \ell(\mathbf{w}^*)) = \sum_{j=1}^t (\ell(\mathbf{w}_j) - \ell_j(\mathbf{w}_j)) + \sum_{j=1}^t (\ell_j(\mathbf{w}^*) - \ell(\mathbf{w}^*)) + \sum_{j=1}^t (\ell_j(\mathbf{w}_j) - \ell_j(\mathbf{w}^*)).$$

From the convexity of $\ell_j$, the term $\sum_{j=1}^{t}(\ell_j(\mathbf{w}_j) - \ell_j(\mathbf{w}^*))$ can be bounded as

$$\sum_{j=1}^{t}(\ell_j(\mathbf{w}_j) - \ell_j(\mathbf{w}^*)) \le \sum_{j=1}^{t} \mathbf{g}_j^\top(\mathbf{w}_j - \mathbf{w}^*) = \sum_{j=1}^{t} \hat{\mathbf{g}}_j^\top(\mathbf{w}_j - \mathbf{w}^*) + \sum_{j=1}^{t}(\mathbf{g}_j - \hat{\mathbf{g}}_j)^\top(\mathbf{w}_j - \mathbf{w}^*).$$

Summarizing the above inequalities, $\sum_{j=1}^{t} \|L(\mathbf{w}_j - \mathbf{w}^*)\|^2$ is bounded by the sum of (i) $\sum_{j=1}^{t} \hat{\mathbf{g}}_j^\top(\mathbf{w}_j - \mathbf{w}^*)$, (ii) $\sum_{j=1}^{t}(\mathbf{g}_j - \hat{\mathbf{g}}_j)^\top(\mathbf{w}_j - \mathbf{w}^*)$, (iii) $\sum_{j=1}^{t}(\ell(\mathbf{w}_j) - \ell_j(\mathbf{w}_j))$, and (iv) $\sum_{j=1}^{t}(\ell_j(\mathbf{w}^*) - \ell(\mathbf{w}^*))$. Next, let us construct bounds for each term.

From Lemma 9, (i) $\sum_{j=1}^{t} \hat{\mathbf{g}}_j^\top(\mathbf{w}_j - \mathbf{w}^*)$ can be bounded as

$$\sum_{j=1}^{t} \hat{\mathbf{g}}_j^\top(\mathbf{w}_j - \mathbf{w}^*) \le \sum_{j=1}^{t} \frac{G^2}{\lambda_j} + \frac{\lambda_{t+1}}{2}$$

with probability 1. Next, consider (ii). Denote the $j$-th input data $(\mathbf{x}_j, y_j)$ by $z_j$. Since we have

$$\mathbf{E}[(\mathbf{g}_j - \hat{\mathbf{g}}_j)(\mathbf{w}_j - \mathbf{w}^*)|z_1, \ldots, z_{j-1}, S_1, \ldots, S_{j-1}] = 0$$

and $|(\mathbf{g}_j - \hat{\mathbf{g}}_j)(\mathbf{w}_j - \mathbf{w}^*)| \le 2G$, we can apply the Azuma–Hoeffding inequality to (ii) to obtain that

$$\mathrm{Prob}[\sum_{j=1}^{t}(\mathbf{g}_j - \hat{\mathbf{g}}_j)^\top(\mathbf{w}_j - \mathbf{w}^*) \ge \delta] \le \exp(\frac{-\delta^2}{8tG^2}).$$

The value of (iii) can also be bounded by using the Azuma–Hoeffding inequality, since we have

$$\mathbf{E}[\ell(\mathbf{w}_j) - \ell_j(\mathbf{w}_j)|z_1, \ldots, z_{j-1}, S_1, \ldots, S_{j-1}] = 0,$$

and $|\ell(\mathbf{w}_j) - \ell_j(\mathbf{w}_j)| \le Q$. Accordingly, we obtain

$$\mathrm{Prob}[\sum_{j=1}^{t} \ell(\mathbf{w}_j) - \ell_j(\mathbf{w}_j) \ge \delta] \le \exp(\frac{-\delta^2}{2tR^Q}).$$

Similarly, since $\{\ell_j(\mathbf{w}^*) - \ell(\mathbf{w}^*)\}$ are independent random variables such that $\mathbf{E}[\ell_j(\mathbf{w}^*) - \ell(\mathbf{w}^*)] = 0$ and $|\ell_j(\mathbf{w}^*) - \ell(\mathbf{w}^*)| \le Q$ the value of (iv) can be bounded by using Hoeffding's inequality, as follows:

$$\mathrm{Prob}[\sum_{j=1}^{t} \ell_j(\mathbf{w}^*) - \ell(\mathbf{w}^*) \ge \delta] \le \exp(\frac{-\delta^2}{2tQ^2}).$$

Summarizing the above inequalities, we have

$$\mathrm{Prob}\left[\sum_{j=1}^{t} \|L(\mathbf{w}_j - \mathbf{w}^*)\|^2 \ge \sum_{j=1}^{t} \frac{G^2}{\lambda_j} + \frac{\lambda_{t+1}}{2} + 3\delta\right] \le 2\exp(\frac{-\delta^2}{2tQ^2}) + \exp(\frac{-\delta^2}{8tG^2}) \le 3\exp(\frac{-\delta^2}{Ct}).$$

By substituting this for $\lambda_j = 2G\sqrt{j}$, we obtain (20). $\qquad\square$

We note that the probability claimed in Proposition 11 is over the randomness of both the examples and Algorithm 1.

**Computing $\mathbf{w}_s$, $\bar{\mathbf{w}}_s$, and $\hat{\mathbf{g}}_s$.** As noted above, $\bar{\mathbf{w}}_s$ is defined as the average of $\mathbf{w}_1, \ldots, \mathbf{w}_s$ computed as in Algorithm 1 applied to the examples $\{(\mathbf{x}_{t_1}, y_{t_1}), \ldots, (\mathbf{x}_{t_s}, y_{t_s})\}$, setting $k_1 \le k - 2$. Recall that Algorithm 1 computes $\mathbf{w}_s$ from $\hat{\mathbf{g}}_1, \ldots, \hat{\mathbf{g}}_{s-1}$ using (5), and computes $\hat{\mathbf{g}}_j$ from $\mathbf{w}_{t_j}$ and $\{(D_{t_1}\mathbf{x}_{t_1}, y_{t_1}), \ldots, (D_{t_j}\mathbf{x}_{t_j}, y_{t_j})\}$ using (6) for any $j \in [s]$. We use $D_{t_j}$ defined from $S_{t_j}$ in Algorithm 2 instead of Algorithm 1.

For convenience, we define $\bar{\mathbf{w}}_0$ as the zero vector.

**Computing $S_t$.** Let $s$ be the largest number such that $t_s \leq t$. Then $S_t$ is defined as the set of $k$ largest features with respect to $\bar{\mathbf{w}}_s$. Note that $S_t$ is the same for all $t$ with $t_s \leq t < t_{s+1}$. In the following, we show from Proposition 11 that $S_t$ contains $S^*$ with high probability.

**Lemma 12.** *If $w_i^{*2}\sigma_d^2 - 8Gs^{-\frac{1}{2}} \geq 0$, the following holds for any $i \in S^*$ and $t = t_s, \ldots, t_{s+1} - 1$:*

$$\mathrm{Prob}[i \notin S_t] \leq 3 \exp\left(-\frac{C^2 s}{4608}(w_i^{*2}\sigma_d^2 - \frac{32}{C}s^{-\frac{1}{2}})^2\right).$$

*Proof.* If a feature $i$ satisfies $i \in S^* \setminus S_t$, it holds that $\|\bar{\mathbf{w}}_s - \mathbf{w}^*\|^2 \geq w_i^{*2}/2$, which can be confirmed as follows. Since $|S^*| = |S_t|$, $i \in S^* \setminus S_t$ means that there exist $j \in S_t \setminus S^*$ such that $|\bar{w}_{si}| < |\bar{w}_{sj}|$, which implies that $\|\bar{\mathbf{w}}_s - \mathbf{w}^*\|^2 \geq (\bar{w}_{si} - w_i^*)^2 + (\bar{w}_{sj} - w_j^*)^2 = (\bar{w}_{si} - w_i^*)^2 + \bar{w}_{sj}^2 > (\bar{w}_{si} - w_i^*)^2 + \bar{w}_{si}^2 \geq w_i^{*2}/2$, where the first equality comes from $j \notin S^*$, the second inequality comes from $|\bar{w}_{si}| < |\bar{w}_{sj}|$, and the third inequality holds for arbitrary $\bar{w}_{si} \in \mathbb{R}$. Hence, we have

$$\mathrm{Prob}[i \notin S_t] \leq \mathrm{Prob}[\|\bar{\mathbf{w}}_s - \mathbf{w}^*\|^2 \geq w_i^{*2}/2]. \tag{21}$$

From Proposition 11, if $\delta := \frac{1}{6}(s\sigma_d^2 w_i^{*2} - 4G\sqrt{s+1}) \geq 0$, we have

$$\mathrm{Prob}[i \notin S_t] \leq 3\exp(\frac{-\delta^2}{Cs})$$

for $i \in S^2$. From the inequality $\delta \geq \frac{s}{6}(\sigma_d^2 w_i^{*2} - 8Gs^{-\frac{1}{2}})$, we obtain Lemma 12.

$\square$

**Computing $\tilde{\mathbf{w}}_t$ and $\tilde{\mathbf{g}}_t$.** We define $\tilde{\mathbf{w}}_1 = 0$. If $t \geq 2$, $\tilde{\mathbf{w}}_t$ is defined as follows. Recall that $D_1\tilde{\mathbf{g}}_1, \ldots, D_{t-1}\tilde{\mathbf{g}}_{t-1}$ are available at the beginning of round $t$. Let $\tilde{\mathbf{h}}_{t-1} = \sum_{j=1}^{t-1} D_j\tilde{\mathbf{g}}_j$. We prepare a sequence $(\tilde{\lambda}_1, \ldots, \tilde{\lambda}_T)$ of non-negative numbers in advance, and $\tilde{\lambda}_t$ is used in round $t$. Then, $\tilde{\mathbf{w}}_t$ is defined by

$$\tilde{\mathbf{w}}_t = \underset{\mathbf{w} \in \mathbb{R}^d, \|\mathbf{w}\| \leq 1}{\arg\min} \left\{ \tilde{\mathbf{h}}_{t-1}^\top \mathbf{w} + \frac{\tilde{\lambda}_t}{2}\|\mathbf{w}\|^2 \right\}. \tag{22}$$

We define $\tilde{\mathbf{g}}_t$ as the gradient of the loss function $\ell_t(\mathbf{w})$ at $\mathbf{w} = D_t\tilde{\mathbf{w}}_t$, i.e.,

$$\tilde{\mathbf{g}}_t = \nabla_\mathbf{w}\ell_t(D_t\tilde{\mathbf{w}}_t) = 2\mathbf{x}_t(\mathbf{x}_t^\top D_t\tilde{\mathbf{w}}_t - y_t). \tag{23}$$

Note that we cannot compute $\tilde{\mathbf{g}}_t$ because all features in $\mathbf{x}_t$ cannot be observed. Nevertheless, we can compute $D_t\tilde{\mathbf{g}}_t$ from available information $D_t\mathbf{x}_t$, $y_t$, and $\tilde{\mathbf{w}}_t$.

**Regret bound of Algorithm 2** We prove that Algorithm 2 achieves $O(\sqrt{dT})$ regret under the independence of features assumption.

**Lemma 13.** *If $\mathbf{w} \in \mathbb{R}^d$ satisfies $\|\mathbf{w}\| \leq 1$, then we have*

$$R_T(\mathbf{w}) \leq \sum_{t=1}^{T} \mathbf{w}^\top (D_t - I)\tilde{\mathbf{g}}_t + \sum_{t=1}^{T} \frac{\|D_t\tilde{\mathbf{g}}_t\|^2}{\tilde{\lambda}_t} + \frac{\tilde{\lambda}_{T+1}}{2}. \tag{24}$$

*Proof.* Since $\ell_t(\mathbf{w})$ is convex, we have $\ell_t(D_t\tilde{\mathbf{w}}_t) - \ell_t(\mathbf{w}) \leq \tilde{\mathbf{g}}_t^\top(D_t\mathbf{w}_t - \mathbf{w})$. Hence, the regret $R_T(\mathbf{w})$ can be bounded as

$$R_T(\mathbf{w}) \leq \sum_{t=1}^{T} \tilde{\mathbf{g}}_t^\top(D_t\mathbf{w}_t - \mathbf{w}) = \sum_{t=1}^{T} \tilde{\mathbf{g}}_t^\top D_t\mathbf{w}_t - \sum_{t=1}^{T} \tilde{\mathbf{g}}_t^\top\mathbf{w}.$$

From a similar argument to the proof of Lemma 9, we obtain

$$\sum_{t=1}^{T} \tilde{\mathbf{g}}_t^\top D_t\mathbf{w}_t \leq \tilde{\mathbf{h}}_T^\top\mathbf{w} + \sum_{t=1}^{T} \frac{\|D_t\tilde{\mathbf{g}}_t\|^2}{\tilde{\lambda}_t} + \frac{\tilde{\lambda}_{T+1}\|\mathbf{w}\|^2}{2}.$$

By combining the above two inequalities, we obtain (24). $\square$

**Algorithm 2**

---

**Input:** $\{(\mathbf{x}_t, y_t)\} \subseteq \mathbb{R}^d \times \mathbb{R}$, $\{\lambda_s\}$, $\{\tilde{\lambda}_t\} \subseteq \mathbb{R}_{>0}$, $k' \geq 2$ and $k_1 \geq 0$ such that $0 \leq k_1 \leq k' - 2$,
  $J \subseteq \{1, \ldots, T\}$

1: Set $\hat{\mathbf{h}}_0 = 0, \tilde{\mathbf{h}}_0 = 0, \bar{\mathbf{w}}_0 = 0, s = 0$.
2: **for** $t = 1, \ldots, T$ **do**
3:    **if** $t \in J$ **then**
4:       Set $s = s + 1$.
5:       Define $\mathbf{w}_s$ by (5), and $\bar{\mathbf{w}}_s = \bar{\mathbf{w}}_{s-1} + \mathbf{w}_s$.
6:    **end if**
7:    Define $\tilde{\mathbf{w}}_t$ by (22).
8:    Define $S_t$ by $\texttt{Observe}(\bar{\mathbf{w}}_s, k', k_1)$.
9:    Observe $D_t \mathbf{x}_t$ and output $\hat{y}_t := \tilde{\mathbf{w}}_t^\top D_t \mathbf{x}_t$.
10:    Observe $y_t$.
11:    **if** $t \in J$ **then**
12:       Define $\hat{\mathbf{g}}_s$ by (6), and set $\hat{\mathbf{h}}_s = \hat{\mathbf{h}}_{s-1} + \hat{\mathbf{g}}_s$.
13:    **end if**
14:    Compute $D_t \tilde{\mathbf{g}}_t$ ($\tilde{\mathbf{g}}_t$ is defined by (23)).
15:    Set $\tilde{\mathbf{h}}_t = \tilde{\mathbf{h}}_{t-1} + D_t \tilde{\mathbf{g}}_t$.
16: **end for**

---

Because of (24), if $\|\tilde{\mathbf{g}}_t\| = O(1)$ holds and we set $\tilde{\lambda}_t = \Theta(\sqrt{t})$, we have

$$\mathbf{E}[R_T(\mathbf{w}^*)] = O\left(\sum_{t=1}^T \|(D_t - I)\mathbf{w}^*\| + \sqrt{T}\right).$$

From Lemma 12, we can prove that $S_t$ satisfies $\sum_{t=1}^T \sum_{j \in S^*} \text{Prob}[j \notin S_t] = O(\sqrt{T})$. Combining these two facts, we obtain $O(\sqrt{T})$ regret. A more precise statement is given in the following theorem.

**Proof of Theorem 6**

*Proof.* From Lemma 13 and that $\|D_t \tilde{\mathbf{g}}_t\|^2 \leq 4$ and $\tilde{\lambda}_t = 8\sqrt{t}$, we have

$$R_T(\mathbf{w}^*) \leq \sum_{t=1}^T \mathbf{w}^{*\top}(D_t - I)\tilde{\mathbf{g}}_t + \sum_{t=1}^T \frac{\|D_t \tilde{\mathbf{g}}_t\|^2}{\tilde{\lambda}_t} + \frac{\tilde{\lambda}_{T+1}}{2}$$

$$= \sum_{t=1}^T \mathbf{w}^{*\top}(D_t - I)\tilde{\mathbf{g}}_t + 8\sqrt{T+1}.$$

Further, since $((D_t - I)\mathbf{w}^*)_i = -w_i^*$ if $i \in S^* \setminus S_t$ and $((D_t - I)\mathbf{w}^*)_i = 0$ otherwise, the first term can be bounded as

$$\sum_{t=1}^T \mathbf{w}^{*\top}(D_t - I)\tilde{\mathbf{g}}_t = -\sum_{t=1}^T \sum_{i \in S^* \setminus S_t} w_i^* \tilde{g}_{ti} \leq 4 \sum_{t=1}^T \sum_{i \in S^* \setminus S_t} |w_i^*|,$$

where the last inequality comes from $\|\tilde{g}_t\| \leq 4$. From the above two inequalities, by taking the expectation, we obtain

$$\mathbf{E}[R_T(\mathbf{w}^*)] \leq 4 \sum_{t=1}^T \sum_{i \in S^*} |w_i^*| \text{Prob}[i \notin S_t] + 8\sqrt{T+1}. \tag{25}$$

Next, we give a upper bound on $\sum_{t=1}^T \text{Prob}[i \notin S_t]$ for $i \in S^*$ by using Lemma 12. Define $\gamma(s) = \frac{1}{2}\sigma_d^2 w_i^{*2} - 8Gs^{-\frac{1}{2}}$. If $s$ is large enough so that $\gamma(s) \geq 0$, i.e., if $s \geq 256 \frac{G^2}{\sigma_d^4 w_i^{*4}} =: \kappa_i$, then we

have $\sigma_d^2 w_i^{*2} - 8Gs^{-\frac{1}{2}} \geq \frac{1}{2}\sigma_d^2 w_i^{*2}$. From Lemma 12, then, we have $\mathrm{Prob}[i \in S_t] \leq 3\exp(-\frac{s\sigma_d^2 w_i^{*2}}{144C})$. Thus, we have

$$
\begin{aligned}
\sum_{t=1}^{T} \mathrm{Prob}[i \notin S_t] &= \sum_{t \in J \cap [T]} \mathrm{Prob}[i \notin S_t] + \sum_{t \in [T] \setminus J} \mathrm{Prob}[i \notin S_t] \\
&\leq \sqrt{T} + \sum_{t \in [T] \setminus J} \mathrm{Prob}[i \notin S_t] \\
&= \sqrt{T} + \sum_{t \in [T] \setminus J, \sqrt{t}-1 \leq \kappa_i} \mathrm{Prob}[i \notin S_t] + \sum_{t \in [T] \setminus J, \sqrt{t}-1 > \kappa_i} \mathrm{Prob}[i \notin S_t] \\
&\leq \sqrt{T} + (\kappa_i + 1)^2 + \sum_{t \in [T] \setminus J, \sqrt{t}-1 > \kappa_i} \mathrm{Prob}[i \notin S_t], \quad (26)
\end{aligned}
$$

where the first inequality comes from $|[T] \cup J|$, and the second inequality comes from $|\{t \geq 1 \mid \sqrt{t}-1 \leq \kappa_i\}| \leq (\kappa_i + 1)^2$. Note that, since $s \geq \sqrt{t}-1$ holds in each step $t$, $\sqrt{t}-1 > \kappa_i$ implies that $s > \kappa_i$ and, hence, $\mathrm{Prob}[i \notin S_t] \leq 3\exp(-\frac{s\sigma_d^2 w_i^{*2}}{144C}) \leq 3\exp(-\frac{(\sqrt{t}-1)\sigma_d^2 w_i^{*2}}{144C})$ holds. From this, the last term of (26) can be bounded as

$$
\begin{aligned}
\sum_{t \in [T] \setminus J, \sqrt{t}-1 > \kappa_i} \mathrm{Prob}[i \notin S_t] &= \sum_{t \in [T] \setminus J, \sqrt{t}-1 > \kappa_i, t \leq \sqrt{T}} \mathrm{Prob}[i \notin S_t] + \sum_{t \in [T] \setminus J, \sqrt{t}-1 > \kappa_i, t > \sqrt{T}} \mathrm{Prob}[i \notin S_t] \\
&\leq \sqrt{T} + \sum_{t \in [T] \setminus J, \sqrt{t}-1 > \kappa_i, t > \sqrt{T}} 3\exp\left(-\frac{(\sqrt{t}-1)\sigma_d^2 w_i^{*2}}{144C}\right) \\
&\leq \sqrt{T} + 3T\exp\left(\frac{-(T^{\frac{1}{4}}-1)\sigma_d^2 w_i^{*2}}{144C}\right)
\end{aligned}
$$

By combining the above two inequalities, we obtain

$$
\sum_{t=1}^{T} \mathrm{Prob}[i \notin S_t] \leq 2\sqrt{T} + (\kappa_i + 1)^2 + 3T\exp\left(\frac{-(T^{\frac{1}{4}}-1)\sigma_d^2 w_i^{*2}}{144C}\right).
$$

By substituting this inequality into (25), we have

$$
\mathbf{E}[R_T(\mathbf{w}^*)] \leq 8\sum_{i \in S^*} |w_i^*|\sqrt{T} + 8\sqrt{T+1} + 4\sum_{i \in S^*} |w_i^*|\left((\kappa_i + 1)^2 + 3T\exp\left(\frac{-(T^{\frac{1}{4}}-1)\sigma_d^2 w_i^{*2}}{144C}\right)\right).
$$

The first term of the right-hand side can be bounded as $4\sum_{i \in S^*} |w_i^*|\sqrt{T} = \|\mathbf{w}^*\|_1 4T \leq \sqrt{d}4\sqrt{T+1}$ because $\|\mathbf{w}^*\| \leq 1$. Further, substituting $\kappa_i = 256\frac{G^2}{\sigma_d^4 w_i^{*4}}$, we obtain Theorem 6. $\square$

## E Details of Algorithm 3

**Computing $\mathbf{w}_s$ and $\bar{\mathbf{w}}_s$.** Let $\{\lambda_j\}, \{\eta_j\} \subseteq \mathbb{R}_{>0}$ be positive monotone increasing sequences. Denote $\zeta_j = \eta_j - \eta_{j-1}$ for $j > 1$ and $\zeta_1 = \eta_1$. Then, we have $\zeta_j > 0$ and $\eta_s = \sum_{j=1}^{s} \zeta_j$.

Recall that $\hat{\mathbf{h}}_{s-1} = \sum_{j=1}^{s-1} \hat{\mathbf{g}}_j$. We define $\mathbf{w}_s$ and $\bar{\mathbf{w}}_s$ by

$$
\mathbf{w}_s = \arg\min_{\mathbf{w} \in \mathbb{R}^d, \|\mathbf{w}\| \leq 1} \left\{ \hat{\mathbf{h}}_{s-1}^\top \mathbf{w} + \frac{\lambda_s}{2}\|\mathbf{w}\|^2 + \eta_s\|\mathbf{w}\|_1 \right\} = -\frac{1}{\max\{\lambda_t, \|\hat{\mathbf{h}}_{t-1}\|/1\}}\hat{\mathbf{h}}_t,
$$

and $\bar{\mathbf{w}}_s = \sum_{j=1}^{s} \zeta_j \mathbf{w}_j / \eta_s$. Then, $\bar{\mathbf{w}}_s$ gets close to $\mathbf{w}^*$ with high probability.

**Lemma 14.** *Let $G$ and $C$ be as defined in Section B. Set $\lambda_j = \frac{8}{C}\sqrt{j}$ and $\zeta_j = \phi_0\sqrt{4/(Ck)}j^{-\frac{1}{4}}$. Under the compatibility assumption, for arbitrary $\delta > 0$, $4\phi_0(s^{\frac{3}{4}}-1)\sqrt{\frac{4}{Ck}}\|\bar{\mathbf{w}}_s - \mathbf{w}^*\|_1 \leq \frac{144}{C}\sqrt{s+1} + 27\delta$ holds with probability at least $1 - 3\exp(\frac{-C^2\delta^2}{128s})$ over the randomness of both the examples and the algorithm.*

*Proof.* From the convexity of the triangle inequality of $\ell_1$ norm, we have

$$\eta_t \|\bar{\mathbf{w}}_t - \mathbf{w}^*\|_1 = \|\sum_{j=1}^{t} \zeta_j(\mathbf{w}_t - \mathbf{w}^*)\|_1 \le \sum_{j=1}^{t} \zeta_j \|\mathbf{w}_t - \mathbf{w}^*\|_1.$$

Hence, it suffices to give a bound on $\sum_{j=1}^{t} \zeta_j \|\mathbf{w}_t - \mathbf{w}^*\|_1$. Define $\gamma_j = \|L(\mathbf{w}_j - \mathbf{w}^*)\|^2 + \zeta_j(\|\mathbf{w}_j\|_1 - \|\mathbf{w}^*\|_1)$, and we shall show the following bound on $\zeta_j \|\mathbf{w}_s - \mathbf{w}^*\|_1$:

$$\zeta_j \|\mathbf{w}_j - \mathbf{w}^*\|_1 \le 3k\zeta_j^2/\phi_0^2 + 3\gamma_j$$

under the assumption of the compatibility condition, i.e., $\|\mathbf{w}_{[d]\setminus S^*}\|_1 \le 2\|\mathbf{w}_{S^*}\|_1 \implies \phi_0^2\|\mathbf{w}_{S^*}\|_1^2 \le k\|L\mathbf{w}\|^2$. In the following, we use the notation $\Delta = \mathbf{w}_j - \mathbf{w}^*$ for convenience. Then, we have

$$\begin{aligned}
\gamma_j &= \|L\Delta\|^2 + \zeta_j(\|\mathbf{w}_j|_{S^*}\|_1 + \|\mathbf{w}_j|_{S^{*c}}\|_1 - \|\mathbf{w}^*|_{S^*}\|_1) \\
&\ge \|L\Delta\|^2 + \zeta_j(\|\Delta|_{S^{*c}}\|_1 - \|\Delta|_{S^*}\|_1) \\
&\ge \zeta_j(\|\Delta|_{S^{*c}}\|_1 - \|\Delta|_{S^*}\|_1). \qquad (27)
\end{aligned}$$

We will bound $\Delta$ by considering the following two cases: (i) $\gamma_j \le \zeta_j\|\Delta|_{S^*}\|_1$ and (ii) $\gamma_j > \zeta_j\|\Delta|_{S^*}\|_1$.

Case (i) $\gamma_j \le \zeta_j\|\Delta|_{S^*}\|_1$:
From (27) and $\gamma_j \le \zeta_j\|\Delta|_{S^*}\|_1$, we have

$$\zeta_j\|\Delta|_{S^{*c}}\|_1 \le \zeta_j\|\Delta|_{S^*}\|_1 + \gamma_j \le 2\zeta_j\|\Delta|_{S^*}\|_1.$$

From the compatibility condition, we have $\phi_0^2\|\Delta|_{S^*}\|_1^2 \le k\|L\Delta\|^2$. Hence, we have

$$\begin{aligned}
\zeta_j\|\Delta\|_1 &= \zeta_j\|\Delta|_{S^*}\|_1 + \zeta_j\|\Delta|_{S^{*c}}\|_1 \\
&\le 2\zeta_j\|\Delta|_{S^*}\|_1 + \gamma_j - \|L\Delta_j\|^2 \\
&\le 2\zeta_j\|\Delta|_{S^*}\|_1 + \gamma_j - \phi_0^2\|\Delta|_{S^*}\|_1^2/k \\
&\le k\zeta_j^2/\phi_0^2 + \gamma_j \le 3k\zeta_j^2/\phi_0^2 + 3\gamma_j,
\end{aligned}$$

where the first, second, and third inequalities come from (27), the compatibility condition, and completing the square.

Case (ii) $\gamma_j > \zeta_j\|\Delta|_{S^*}\|_1$:
From (27) and $\gamma_j \le \zeta_j\|\Delta|_{S^*}\|_1$, we have

$$\zeta_j\|\Delta\|_1 = \zeta_j\|\Delta|_{S^*}\|_1 + \zeta_j\|\Delta|_{S^{*c}}\|_1 \le 2\zeta_j\|\Delta|_{S^*}\|_1 + \gamma_j \le 3\gamma_j.$$

From the argument on cases (i) and (ii), we have $\zeta_j\Delta = \zeta_j\|\mathbf{w}_j - \mathbf{w}^*\|_1 \le 3k\zeta_j^2/\phi_0^2 + 3\gamma_j$. Taking the sum over $j = 1, 2, \ldots, t$, we have

$$\sum_{j=1}^{t} \zeta_j\|\mathbf{w}_j - \mathbf{w}^*\|_1 \le \frac{3k}{\phi_0^2} \sum_{j=1}^{s} \zeta_j^2 + 3\sum_{j=1}^{s} \gamma_j.$$

Here, from Lemma 10, for all $\delta > 0$, the value $\sum_{j=1}^{s} \gamma_j$ can be bounded as

$$\begin{aligned}
\sum_{j=1}^{s} \gamma_j &= \sum_{j=1}^{s} (\|L(\mathbf{w}_j - \mathbf{w}^*)\|^2 + \zeta_j(\|\mathbf{w}_j\|_1 - \|\mathbf{w}^*\|_1)) \\
&\le \sum_{j=1}^{s} (\hat{\mathbf{g}}_j^\top(\mathbf{w}_j - \mathbf{w}^*) + \zeta_j(\|\mathbf{w}_j\|_1 - \|\mathbf{w}^*\|_1)) + 3\delta
\end{aligned}$$

with probability at least $1 - 3\exp\left(\frac{-\delta^2}{Cs}\right)$. Further, from Lemma 9 the right-most side can be bounded as

$$\begin{aligned}
&\sum_{j=1}^{s} (\hat{\mathbf{g}}_j^\top(\mathbf{w}_j - \mathbf{w}^*) + \zeta_j(\|\mathbf{w}_j\|_1 - \|\mathbf{w}^*\|_1)) + 3\delta \\
&\le \sum_{j=1}^{s} \frac{1}{\lambda_j}\|\hat{\mathbf{g}}_j\|^2 + \frac{1\lambda_{s+1}}{2} + 3\delta \le \sum_{j=1}^{s} \frac{16}{\lambda_j} + \frac{1\lambda_{s+1}}{2} + 3\delta
\end{aligned}$$

with probability one. Summarizing the above argument, it holds that

$$\eta_s \|\bar{\mathbf{w}}_s - \mathbf{w}^*\| \leq \frac{3k}{\phi_0^2} \sum_{j=1}^{s} \zeta_j^2 + 3 \sum_{j=1}^{s} \frac{16}{\lambda_j} + \frac{3\lambda_{s+1}}{2} + 9\delta$$

with probability at least $1 - 3\exp\left(\frac{-\delta^2}{Cs}\right)$. By assigning $\lambda_j = 2G\sqrt{j}$ and $\zeta_j = \phi_0\sqrt{G/k}j^{-\frac{1}{4}}$, we obtain Lemma 14.

$\square$

**Regret bound of Algorithm 3** We prove that Algorithm 3 achieves $O(\sqrt{dT})$ regret assuming the compatibility condition. Recall that $S_t$ is the set of the $k$ largest features with respect to $\bar{\mathbf{w}}_s$, From Lemma 14, $S_t$ contains $S^*$ with high probability as follows.

**Lemma 15.** *Let $G$ and $C$ be constants defined as in Section B. Let $\{\lambda_j\}$ and $\{\zeta_j\}$ be sequences defined as in Lemma 14. For any $i \in S^*$ and $t = t_s, \ldots, t_{s+1}-1$, if $\delta := \frac{1}{27}(4\phi_0(s^{\frac{3}{4}}-1)|w_i^*|\sqrt{\frac{4}{Ck}} - \frac{144}{C}\sqrt{s+1}) > 0$, then we have $\mathrm{Prob}[i \notin S_t] \leq 3\exp(\frac{-C^2\delta^2}{128s})$.*

*Proof.* If a feature $i$ satisfies $i \in S^* \setminus S_t$, it holds that $\|\bar{\mathbf{w}}_s - \mathbf{w}^*\|_1 \geq |w_i^*|$, which can be confirmed as follows. Since $|S^*| = |S_t|$, $i \in S^* \setminus S_t$ means that there exist $j \in S_t \setminus S^*$ such that $|\bar{w}_{sj}| > |\bar{w}_{si}|$, which implies that $\|\bar{\mathbf{w}}_s - \mathbf{w}^*\|_1 \geq |\bar{w}_{si} - w_i^*| + |\bar{w}_{sj} - w_j^*| = |\bar{w}_{si} - w_i^*| + |\bar{w}_{sj}| \geq |\bar{w}_{si} - w_i^*| + |\bar{w}_{si}| \geq |w_i^*|$, where the first equality comes from $j \notin S^*$, the second inequality comes from $|\bar{w}_{sj}| > |\bar{w}_{si}|$, and the third is the triangle inequality. Hence, we have

$$\mathrm{Prob}[i \notin S_t] \leq \mathrm{Prob}[\|\bar{\mathbf{w}}_s - \mathbf{w}^*\|_1 \geq |w_i^*|].$$

From Lemma 14, if $\delta := \frac{1}{27}(4\phi_0(s^{\frac{3}{4}} - 1)\sqrt{G/k}|w_i^*| - 36G\sqrt{s+1}) \geq 0$, we have

$$\mathrm{Prob}[\|\bar{\mathbf{w}}_s - \mathbf{w}^*\|_1 \geq |w_i^*|] \leq 3\exp(\frac{-\delta^2}{Cs})$$

for $i \in S^*$. The above two inequalities yield Lemma 15. $\square$

**Proof of Theorem 7**

*Proof.* The outline of the proof is similar to that of Theorem 6. From Lemma 13, we obtain

$$\mathbf{E}[R_T(\mathbf{w}^*)] \leq 4 \sum_{t=1}^{T} \sum_{i \in S^*} |w_i^*| \mathrm{Prob}[i \notin S_t] + 8\sqrt{T+1}, \tag{28}$$

in a similar way to the proof of Theorem 6.

Next, we give a upper bound on $\sum_{t=1}^{T} \mathrm{Prob}[i \notin S_t]$ for $i \in S^*$ by using Lemma 15. Define $\gamma(s) = 2|w_i^*|\phi_0\sqrt{G/k} - 4|w_i^*|\phi_0\sqrt{G/k}s^{-\frac{3}{4}} - 36Gs^{-\frac{3}{4}}\sqrt{s+1}$. If $s$ is large enough so that $\gamma(s) \geq 0$, from Lemma 12, we have $\mathrm{Prob}[i \in S_t] \leq 3\exp(-\frac{4\sqrt{s}|w_i^*|^2\phi_0^2 G}{27^2 Ck})$. Note that if $s \geq \frac{4\cdot36^4 G^2 k^2}{w_i^{*4}\phi_0^4} =: \kappa_i$, it holds that $\gamma(s) > 0$, from the definition of $\gamma(s)$. Further, since $s \geq \sqrt{t} - 1$ in each step $t$, if $\sqrt{t} - 1 \geq \kappa_i$, we have $\mathrm{Prob}[i \in S_t] \leq 3\exp(-\frac{4\sqrt{\sqrt{t}-1}|w_i^*|^2\phi_0^2 G}{27^2 Ck})$. From this property of $S_t$, by a similar argument to the proof of Theorem 6, we have

$$\sum_{t=1}^{T} \mathrm{Prob}[i \notin S_t] \leq 2\sqrt{T} + (\kappa_i + 1)^2 + 3T\exp(-\frac{4\sqrt{T^{\frac{1}{4}} - 1}|w_i^*|^2\phi_0^2 G}{27^2 Ck}).$$

By combining this inequality with (28), we obtain

$$\mathbf{E}[R_T(\mathbf{w}^*)] \leq 8 \sum_{i \in S^*} |w_i^*|\sqrt{T} + 8\sqrt{T+1}$$

$$+ \sum_{i \in S^*} |w_i^*|((\kappa_i + 1)^2 + 3T\exp(-\frac{4\sqrt{T^{\frac{1}{4}} - 1}|w_i^*|^2\phi_0^2 G}{27^2 Ck})).$$

The first term of the right-hand side can be bounded as $4\sum_{i \in S^*} |w_i^*|\sqrt{T} = \|\mathbf{w}^*\|_1 4T \leq \sqrt{d}8\sqrt{T+1}$ because $\|\mathbf{w}^*\| \leq 1$. Further, substituting $\kappa_i = \frac{4\cdot36^4 G^2 k^2}{w_i^{*4}\phi_0^4}$, we obtain Theorem 6. $\square$

# F   More Experiments

In this section, we provide supplementary descriptions of our experiments.

**Experimental environment.**   The experiments were performed on a server with Intel Xeon E5-2680 v3 CPUs. All algorithms are implemented in Python.

**The generation procedure of synthetic datasets.**   We first create the ground-truth weight vector $\mathbf{w}_{\text{true}}$ by choosing a set $S_k$ of $k$ features from $[d]$ uniformly at random, and setting $w_{\text{true},i} \in [-1, 1]$ for $i \in S_k$ and $w_{\text{true},i} = 0$ otherwise. For each $t \in T$, we generate $\mathbf{x}_t$ by sampling $x_{t,i}$ $(i \in [d])$ from $\mathcal{N}(0, 1)$ and set $y_t = \mathbf{w}_{\text{true}}^\top \mathbf{x}_t + 0.5z$, where $z$ is sampled from $\mathcal{N}(0, 1)$.

**Preprocessing of CT-slice datasets.**   We deal with features that are outside of an image as those having a value of zero. The sequence of the dataset is randomly shuffled to avoid the effects of biased sequences of images. The maximum number of positive features in one image is 165, the minimum is 9, and the average is 73. Thus, we set $k$ to be 60, 70, or 90 in this paper.

**Results on synthetic datasets.**   Figures 4, 5 show typical results for some instance with $(d, k, k') = (20, 5, 7)$. We remark that Figure 1 in the main body plots the regrets for only the first 5000 iterations. We observe that our algorithms achieve small regrets at the end of iterations. Figure 5 indicates that the increase in the regrets of our algorithms is smaller than baseline algorithms for large $T$. We remark that FKK is executed for 5000 iterations because the run time is too expensive compared with the others. However, Figure 1, which plots regrets for the first 5000 iterations, shows that the increase in the regrets for FKK is similar to greedy and uniform-random. Thus, we can expect that our algorithms perform much better than all baseline algorithms.

Figure 4: The regrets for a synthetic instance with $(d, k, k') = (20, 5, 7)$.

Figure 5: The log-log plots of the regrets for a synthetic instance with $(d, k, k') = (20, 5, 7)$.

We compute the averages of final regrets and execution times for each combination of $d \in \{10, 20, 50, 100\}$, $k \in \{2, 5, 10\}$, and $k' \in \{k + 2, k + 5, k + 10\}$. For each combination of $(d, k, k')$, we executed all algorithms on five instances with $T = 5000$. Tables 4 and 5 summarize the average regrets and execution times. In the tables, "T" denotes that we do not execute FKK due to the expensive run time. We observe that Algorithm 1 performs best for almost all cases. Our algorithms, greedy, and uniform-random can all process each round about three times as fast as AELR. Thus, we can say that our algorithms outperform the others in terms of both regrets and execution times.

**Results on CT-slice datasets.**   We present results with $(k, k') = \{(60, 70), (70, 80), (90, 95)\}$ in Figures 6, 7, and 8, respectively. We observe that Algorithms 2 and 3 perform best. The increases in the regrets of our algorithms are much smaller than those of uniform-random and AELR. Greedy performs well at the beginning, but the performance degrades at the end.

Table 4: Average regrets $R_T/10^2$ (A1 = Algorithm 1, A2 = Algorithm 2, A3 = Algorithm 3, G = greedy, U = uniform random).

| $(d,k,k')$ | A1 | A2 | A3 | G | U | AELR | FKK |
|---|---|---|---|---|---|---|---|
| (10,2,4) | **1.53** | 2.38 | 3.60 | 33.28 | 25.73 | 60.76 | 24.05 |
| (10,2,7) | **1.45** | 4.73 | 1.84 | 39.62 | 16.45 | 66.60 | 5.92 |
| (10,5,7) | **3.35** | 46.18 | 16.74 | 79.38 | 44.46 | 145.87 | 57.89 |
| (20,2,4) | **6.03** | 7.31 | 13.63 | 16.52 | 14.89 | 37.15 | 14.90 |
| (20,2,7) | **1.03** | 5.99 | 5.03 | 8.13 | 6.07 | 17.16 | 6.65 |
| (20,2,12) | **0.81** | 3.11 | 3.13 | 8.18 | 4.47 | 16.49 | 6.46 |
| (20,5,7) | **15.45** | 83.53 | 49.58 | 49.37 | 80.55 | 138.05 | 83.50 |
| (20,5,10) | **6.10** | 43.15 | 17.20 | 16.18 | 65.08 | 140.03 | T |
| (20,5,15) | **4.00** | 9.81 | 8.40 | 7.93 | 34.23 | 140.12 | T |
| (20,10,12) | **15.00** | 40.86 | 24.76 | 21.38 | 104.63 | 254.52 | T |
| (20,10,15) | **6.79** | 5.14 | 9.97 | 10.08 | 71.74 | 283.76 | T |
| (100,2,7) | **41.09** | 42.13 | 49.67 | 49.78 | 47.42 | 83.70 | T |
| (100,2,12) | **25.90** | 51.34 | 50.13 | 50.66 | 46.82 | 84.11 | T |
| (100,5,7) | 133.54 | 118.23 | 119.17 | 119.51 | **113.17** | 214.84 | T |
| (100,5,10) | **74.06** | 118.95 | 106.78 | 120.12 | 112.51 | 217.52 | T |
| (100,5,15) | **62.45** | 104.57 | 84.52 | 122.71 | 111.60 | 211.22 | T |
| (100,10,12) | 295.95 | 225.06 | **203.04** | 227.81 | 214.09 | 366.71 | T |
| (100,10,15) | 159.95 | 213.20 | **159.35** | 198.82 | 212.39 | 423.73 | T |
| (100,10,20) | **85.89** | 225.77 | 199.80 | 193.25 | 207.16 | 372.67 | T |

Table 5: Average run time [s] (A1 = Algorithm 1, A2 = Algorithm 2, A3 = Algorithm 3, G = greedy, U = uniform random).

| $(d,k,k')$ | A1 | A2 | A3 | G | U | AELR | FKK |
|---|---|---|---|---|---|---|---|
| (10,2,4) | 0.54 | 0.48 | 0.49 | 0.49 | 0.57 | 4.90 | 3.30 |
| (10,2,7) | 0.81 | 0.50 | 0.51 | 0.63 | 0.89 | 5.03 | 3.50 |
| (10,5,7) | 0.69 | 0.50 | 0.51 | 0.62 | 0.87 | 5.09 | 18.97 |
| (20,2,4) | 0.60 | 0.53 | 0.52 | 0.55 | 0.64 | 6.19 | 13.11 |
| (20,2,7) | 0.87 | 0.54 | 0.59 | 0.66 | 0.93 | 6.20 | 13.01 |
| (20,2,12) | 1.63 | 0.57 | 0.66 | 1.10 | 1.76 | 6.31 | 14.03 |
| (20,5,7) | 0.71 | 0.48 | 0.47 | 0.61 | 0.87 | 5.46 | 1021.51 |
| (20,5,10) | 1.08 | 0.52 | 0.52 | 0.81 | 1.31 | 5.93 | T |
| (20,5,15) | 2.01 | 0.56 | 0.56 | 1.26 | 2.37 | 6.04 | T |
| (20,10,12) | 1.13 | 0.56 | 0.54 | 1.00 | 1.73 | 6.36 | T |
| (20,10,15) | 1.73 | 0.61 | 0.63 | 1.34 | 2.51 | 6.83 | T |
| (100,2,4) | 1.09 | 0.83 | 0.82 | 0.88 | 1.10 | 11.59 | T |
| (100,2,7) | 1.36 | 0.92 | 0.89 | 1.06 | 1.46 | 12.37 | T |
| (100,2,12) | 2.12 | 0.95 | 0.90 | 1.35 | 2.21 | 12.06 | T |
| (100,5,7) | 1.21 | 0.86 | 0.84 | 1.00 | 1.37 | 11.52 | T |
| (100,5,10) | 1.58 | 0.88 | 0.86 | 1.18 | 1.80 | 11.62 | T |
| (100,5,15) | 2.46 | 0.94 | 0.90 | 1.59 | 2.81 | 11.52 | T |
| (100,10,12) | 1.61 | 0.89 | 0.88 | 1.33 | 2.16 | 11.98 | T |
| (100,10,15) | 2.13 | 0.93 | 0.94 | 1.61 | 2.81 | 12.39 | T |
| (100,10,20) | 3.29 | 1.00 | 1.00 | 2.16 | 4.24 | 12.66 | T |

Figure 6: The square loss for CT-slice datasets ($k = 60, k' = 70$).

Figure 7: The square loss for CT-slice datasets ($k = 70, k' = 80$).

Figure 8: The square loss for CT-slice datasets ($k = 90, k' = 95$).