[Reviews · NeurIPS 2017]

Reviewer 1



The paper considers the online sparse regression problem introduced by Kale (COLT'14), in which the online algorithm can only observe a subset of k features of each data point and has to sequentially predict a label based only on this limited observation (it can thus only use a sparse predictor for each prediction). Without further assumptions, this problem has been recently shown to be computationally hard by Foster et al (ALT'16). To circumvent this hardness, the authors assume a stochastic i.i.d. setting, and under two additional distributional assumptions they give efficient algorithms that achieve sqrt(T) regret. The results are not particularly exciting, but they do give a nice counter to the recent computational impossibility of Foster et al, in a setting where the data is i.i.d. and well-specified by a k-sparse vector. One of the main things I was missing in the paper is a proper discussion relating its setup, assumptions and results to the literature on sparse recovery / compressed sensing / sparse linear regression. Currently, the paper only discusses the literature and previous work on attribute efficient learning. In particular, I was not entirely convinced that the main results of the paper are not simple consequences of known results in sparse recovery: since the data is stochastic i.i.d., one could implement a simple follow-the-leader style algorithm by solving the ERM problem at each step; this could be done efficiently by applying a standard sparse recovery/regression algorithm under a RIP condition. It seems that this approach might give sqrt(T)-regret bounds similar to the one established in the paper, perhaps up to log factors, under the authors' Assumption (a) (which is much stronger than RIP: it implies that the data covariance matrix is well conditioned, so the unique linear regression solution is the sparse one). While the authors do consider an additional assumption which is somewhat weaker than RIP (Assumption b), it is again not properly discussed and related to more standard assumptions in sparse recovery (e.g, RIP), and it was not clear to me how reasonable and meaningful this assumption is. Questions / comments: * The "Online" in the title of the paper is somewhat misleading: the setting of the paper is actually a stochastic i.i.d. setting (where the objective is still the regret), whereas "online" typically refers to the worst-case non-statistical online setting. * The description of Alg 1 is quite messy and has to be cleaned up. For example: - line 192, "computing S_t": the entire paragraph was not clear to me. What are the "largest features with respect to w_t"? - line 196: what are i,j? - line 199, "computing g_t": many t subscripts are missing in the text. * Thm 2: Why k_1 is arbitrary? Why not simply set it to k'-2 (which seems to give the best regret bound)? * Can you give a simpler algorithm under Assumption (b) for the easier case where k <= k'+2 (analogous to Alg 1)? Minors / typos: * line 14: the references to [11,10] seems out of context. * line 156, Assumption (a): use an explicit constant in the assumption, rather than O(1); in the analysis you actually implicitly assume this constant is 1. * line 188: "dual average" => "dual averaging". * Lem 3 is central to the paper and should be proven in the main text. * Lem 4 is a standard bound for dual averaging / online gradient descent, and a proper reference should be given. Also, it seems that a simpler, fixed step-size version of the bound suffices for your analysis. * line 226: "form" => "from". * line 246: "Rougly" => "Roughly".

Reviewer 2



While I think the question considered is an important one, I feel the results are weak compared to the strong assumptions made. In particular, I'm most skeptical about 2 points: 1. The regret bound depends on d, the ambient dimension of the feature vectors. This is unacceptable in high-dimensional regimes, where dimension d can be much larger than T. In fact, in order for the regret bound to make sense T has to be much larger than d, which makes the sparsity assumption vacuous. 2. The assumption made in this paper is the strongest possible: random design with least eigenvalue bounded away from below. One cannot make the even stronger beta-min assumption here because it trivializes the problem. Under such strong assumptions one wishes to see the equivalent of fast rate in Lasso (which would translate to log T regret in online learning). This is not the case, as only sqrt{T} slow rate is proved and it is unclear whether it is tight (I would be really surprised if sqrt{T} is optimal, as similar problems in online convex optimization / Lasso shows that with (restricted) strong convexity like the assumptions made in this paper, log T is what to be expected). Minor comments: 3. I suggest emphasizing "with limited/partial observations" in the title and the problem formulation. Online sparse regression refers to the problem where full observations of the features are always available. 4. I do not understand Theorem 1 at all. Assumption (a) implies (b) so why would the problem be difficult with (a)? In fact I suggest not to state (b) as a assumption/condition, but rather a corollary/lemma as it is implied by (a).